# Gamma-Aminobutyric Acid Accumulation Contributes to *Citrus sinensis* Response against ‘*Candidatus* Liberibacter Asiaticus’ via Modulation of Multiple Metabolic Pathways and Redox Status

**DOI:** 10.3390/plants12213753

**Published:** 2023-11-02

**Authors:** Yasser Nehela, Nabil Killiny

**Affiliations:** 1Department of Plant Pathology, Citrus Research and Education Center, University of Florida, 700 Experiment Station Rd., Lake Alfred, FL 33850, USA; yasser.nehela@agr.tanta.edu.eg; 2Department of Agricultural Botany, Faculty of Agriculture, Tanta University, Tanta 31527, Egypt

**Keywords:** huanglongbing, liberibacter, vector-borne diseases, emerging pathogens, citrus, GABA, amino acids, antioxidant

## Abstract

Huanglongbing (HLB; also known as citrus greening) is the most destructive bacterial disease of citrus worldwide with no known sustainable cure yet. Herein, we used non-targeted metabolomics and transcriptomics to prove that γ-aminobutyric acid (GABA) accumulation might influence the homeostasis of several metabolic pathways, as well as antioxidant defense machinery, and their metabolism-related genes. Overall, 41 metabolites were detected in ‘Valencia’ sweet orange (*Citrus sinensis*) leaf extract including 19 proteinogenic amino acids (PAA), 10 organic acids, 5 fatty acids, and 9 other amines (four phenolic amines and three non-PAA). Exogenous GABA application increased most PAA in healthy (except _L_-threonine, _L_-glutamine, _L_-glutamic acid, and _L_-methionine) and ‘*Candidatus* L. asiaticus’-infected citrus plants (with no exception). Moreover, GABA accumulation significantly induced _L-_tryptophan, _L_-phenylalanine, and α-linolenic acid, the main precursors of auxins, salicylic acid (SA), and jasmonic acid (JA), respectively. Furthermore, GABA supplementation upregulated most, if not all, of amino acids, phenolic amines, phytohormone metabolism-related, and GABA shunt-associated genes in both healthy and ‘*Ca*. L. asiaticus’-infected leaves. Moreover, although ‘*Ca*. L. asiaticus’ induced the accumulation of H_2_O_2_ and O_2_^•−^ and generated strong oxidative stress in infected leaves, GABA possibly stimulates the activation of a multilayered antioxidative system to neutralize the deleterious effect of reactive oxygen species (ROS) and maintain redox status within infected leaves. This complex system comprises two major components: (i) the enzymatic antioxidant defense machinery (six *POXs*, four *SODs*, and *CAT*) that serves as the front line in antioxidant defenses, and (ii) the non-enzymatic antioxidant defense machinery (phenolic acids and phenolic amines) that works as a second defense line against ‘*Ca*. L. asiaticus’-induced ROS in citrus infected leaves. Collectively, our findings suggest that GABA might be a promising alternative eco-friendly strategy that helps citrus trees battle HLB particularly, and other diseases in general.

## 1. Introduction

The vector-borne bacterial disease, huanglongbing (HLB; also known as the citrus greening disease), is a devastating citrus disease worldwide [1,2,3,4,5]. The economic impacts of HLB on commercial citrus groves include, but are not limited to, increasing the production costs due to the indirect cost of the management programs of its vector, limitation of fruit production due to crop devastation, and or reduction of the marketable yield per tree, and intensification of the mortality rate of infected trees [3,6,7,8]. Therefore, growers must promptly adopt a comprehensive integrated pest management (IPM) strategy to avoid the rapid collapse of the industry wherever the bacterial pathogen and/or its vector are detected [7].

Although Koch’s postulates of HLB have not been fulfilled yet due to the difficulty in culturing the putative bacterial pathogen, strong evidence indicates that HLB is associated with a fastidious, phloem-limited, α-proteobacterium ‘*Candidatus* Liberibacter spp.’ [3,4,9,10,11]. Based on their geographical distribution and 16S rDNA sequencing, three Liberibacter species were proposed to be associated with HLB including ‘*Ca*. L. asiaticus’ in Asia, Africa, and the Americas, ‘*Ca*. L. americanus’ in Brazil, and ‘*Ca*. L. africanus’ in Africa [3,4,9,10,11]. It is worth mentioning that ‘*Ca.* L. asiaticus’ is the predominant species among the three proposed HLB-associated species, and it can infect most, if not all, citrus cultivars, hybrids, and relatives causing huge economic losses to citrus production worldwide [3,4,12].

The tree-to-tree transmission of ‘*Ca*. Liberibacter spp.’ can occur by graft inoculation; however, they are predominantly transmitted by two citrus psyllid vectors [13]. The African form ‘*Ca*. L. africanus’ is transmitted by the African psyllid *Trioza erytreae* Del Guercio (Hemiptera: Triozidae), while both Asian ‘*Ca*. L. asiaticus’ and the American forms of ‘*Ca*. L. americanus’ are transmitted by the Asian citrus psyllid, *Diaphorina citri* Kuwayama (Hemiptera: Liviidae) [3,13]. Citrus psyllid, *D. citri* particularly, are the most serious phloem-sucking insect of citrus worldwide, especially when bacterial pathogens are also present [13].

Unfortunately, HLB is a fatal disease with no known sustainable cure yet. Therefore, the development of alternative sustainable eco-friendly solutions to help citrus plants struggle with HLB is a desideratum. Citrus resilience through the refining of metabolic-based response(s) is a promising alternative approach that may lead to the battle against HLB. Previously, we proved that citrus plants possess a complex metabolic-based defense system against ‘*Ca*. L. asiaticus’ and its insect vector [5,14,15,16]. This defense system relies on numerous metabolic pathways including, but not limited to, leaf volatile organic compounds (VOCs) [12], amino acids (AA), organic acids (OA), fatty acids (FA) [17], leaf pigments [18], phytohormones [19], polyamines (PAs) [20], melatonin [21], the tricarboxylic acid (TCA)-associated compounds, and non-proteinogenic amino acids (NPAAs) such as γ-aminobutyric acid (GABA) [20].

GABA is ubiquitously distributed NPAA in all plant species that is primarily synthesized from glutamate via a short pathway known as the ‘*GABA shunt*’ [20,22,23,24]. GABA plays multiple key roles in plant growth and development, as well as physiological and molecular processes. For instance, in terms of plant growth and development, GABA is required for pollen tube growth and guidance [25,26], cell elongation [27], cell wall modification [28], and adventitious root growth and formation [29]. Moreover, GABA is involved in several physiological processes including osmotic potential [30], regulation of redox status and cytosolic pH [31,32], modulation of nitrogen (NO_3_^−^-N) uptake, and utilization [33], carbon/nitrogen metabolism [23,28,34], and photosynthesis-associated functions [35,36].

Moreover, GABA is involved directly or indirectly in plant defense responses to abiotic stress [22] such as heat [26], osmotic [30], drought stress [36], salinity [28,32], and salinity–alkalinity stress [31,35]. Additionally, it has been proposed that GABA regulates plant defense against biotic stress [37] such as phytopathogenic fungi [38,39,40], viruses [41,42], and insect herbivores [20,43,44]. Furthermore, GABA was reported to be associated with plant defense against several phytopathogenic bacteria [20,45,46,47,48,49]. Previously, we showed that endogenous GABA levels in citrus leaves were boosted upon ‘*Ca*. L. asiaticus’ infection and/or *D. citri* infestation [17,20]. However, the biochemical and molecular mechanisms behind the GABA-mediated defense responses are poorly understood.

In rice, GABA-mediated defense responses against arsenite toxicity were reported to be associated with the modulation of FA, AA, and polyamine (PAs) biosynthesis [50]. Likewise, in citrus, GABA shunt was suggested to be linked to amino acid and organic acid metabolism during fruit development and post harvest [51,52]. Additionally, exogenous GABA application substantially boosted the endogenous phytohormone content of healthy sweet oranges [53]. Using a non-targeted gas chromatography–mass spectrometry (GC–MS)-based method, we showed that infection with ‘*Ca*. L. asiaticus’ and/or infestation with its insect vector *D. citri* increased the endogenous GABA content in citrus leaves, along with the accumulation of several AA, OA, and FA. [17]. Later, we proved that the GABA shunt is functionally linked with the TCA cycle [20] via three evolutionarily conserved *gab* genes that enable the non-cyclic flux toward succinate, a key intermediate in the TCA cycle, via a GABA shunt in ‘*Ca*. L. asiaticus’-infected citrus [54]. Briefly, GABA is translocated from the cytosol into the mitochondria using mitochondrial GABA permease (*gabP*, aka amino acid permease BAT1-like) where it is catabolized to succinic semialdehyde via the activity of GABA transaminase (*gabT*) then to succinate via the activity of GABA dehydrogenase (aka succinate-semialdehyde dehydrogenase [*SSADH*]) [54].

Moreover, GABA-mediated defense responses are associated with the reduction of reactive oxygen species (ROS) accumulation via activation of antioxidant defense machinery [50,55]. It is worth mentioning that ROS levels, particularly H_2_O_2_, were accumulated to higher levels in different citrus varieties including sour orange, grapefruit, lemon, and sweet orange in response to ‘*Ca*. L. asiaticus’ infection [56]. As a response, proteomic studies showed that several antioxidant enzymes were also up-regulated at the protein level in HLB-affected plants [57,58]. However, to the best of our knowledge, the physiological, biochemical, and molecular relations between GABA and antioxidant enzymes in ‘*Ca*. L. asiaticus’-infected citrus has not been reported previously.

Herein, we hypothesize that GABA contributes to citrus defense responses against ‘*Ca*. L. asiaticus’ via regulation of multiple metabolic pathways and the redox status. We used non-targeted metabolomics and transcriptomic to prove that elevated GABA levels (either due to infection with ‘*Ca*. L. asiaticus’ or exogenous GABA supplementation) might influence the homeostasis of several primary and secondary metabolic pathways including, but not limited to, TCA cycle-related compound, phytohormones, stress-associated amino acids, fatty acids, phenolic amines, and GABA shunt, as well as the transcript levels of their metabolism-related genes and several antioxidant enzymes. Collectively, these metabolic changes might augment the citrus resilience to cope with the HLB disease.

## 2. Results

Using a non-targeted GC-MS-based method, 41 compounds were detected in citrus leaf extract including 19 proteinogenic amino acids (PAA), 10 organic acids, 5 fatty acids (FA), and 9 other amines (4 phenolic amines [PhA] and 3 non-proteinogenic amino acids [NPAA]). Out of these 41 detected metabolites, only _L_-leucine, _L_-aspartic acid, _L_-glutamic acid, _L_-cysteine, _L_-histidine, _L_-tyrosine, and *O*-acetyl serine were not significantly different between healthy and ‘*Ca*. L. asiaticus’-infected ‘Valencia’ sweet orange (*C. sinensis*), neither in non-treated plants nor after GABA supplementation (Table 1).

### 2.1. GABA Alerts the Profile of Proteinogenic Amino Acids (PAA) Levels in Both Healthy and ‘Ca. L. asiaticus’-Infected Plants

Regardless of the impact of GABA application, the endogenous levels of 11 AA (_L_-alanine, _L_-valine, _L_-leucine, _L_-aspartic acid, _L_-serine, _L_-glutamic acid, _L_-methionine, _L_-cysteine, _L_-lysine, _L_-histidine, and _L_-tyrosine) did not change significantly and _L_-glutamine (1294.6 ± 282.2 ng g^−1^ FW) was significantly reduced, whereas the rest of seven AA (glycine, _L_-isoleucine, _L_-threonine, _L_-asparagine, _L_-proline, _L_-phenylalanine, and _L_-tryptophan) were significantly increased upon the infection with ‘*Ca*. L. asiaticus’ (α_adjusted_ ≤ 0.0083; Table 1). Nevertheless, exogenous GABA application did not change AA content in healthy except for _L_-alanine, _L_-isoleucine, which significantly increased (2023.6 ± 301.4 and 150.0 ± 18.1 ng g^−1^ FW, respectively), and _L_-methionine, which markedly decreased (31.4 ± 12.4 ng g^−1^ FW) compared with non-treated control (770.2 ± 311.2, 73.2 ± 14.0, and 129.5 ± 60.5 ng g^−1^ FW, respectively) On the other hand, exogenous GABA application did not change the content of most AA in ‘*Ca*. L. asiaticus’-infected citrus plants except for glycine, _L_-alanine, _L_-valine, _L_-asparagine, _L_-phenylalanine, _L_-lysine, and _L_-tryptophan, which were significantly accumulated upon GABA supplementation (Table 1). The endogenous levels of the most abundant amino acid _L_-proline were significantly increased in ‘*Ca*. L. asiaticus’-infected leaves (33,562.7 ± 1616.1 ng g^−1^ FW) compared with non-treated healthy control (16,371.3 ± 1755.6 ng g^−1^ FW).

It is worth mentioning that GABA supplementation significantly increased the levels of phytohormone precursors amino acids in ‘*Ca*. L. asiaticus’-infected plants, but not in healthy leaves. For instance, _L_-phenylalanine, the precursor of salicylic acid (SA), was increased in ‘*Ca*. L. asiaticus’-infected plants (799.9 ± 75.7 ng g^−1^ FW) compared to non-treated control (326.0 ± 54.9 ng g^−1^ FW). Moreover, GABA application markedly increased the accumulation of _L_-phenylalanine in treated ‘*Ca*. L. asiaticus’-infected leaves (1738.7 ± 445.9 ng g^−1^ FW; *p* = 0.0005), but not in GABA-treated healthy leaves (367.2 ± 67.6 ng g^−1^ FW; *p* = 0.2733) and (Table 1). Additionally, the auxins precursor, _L-_tryptophan, was similarly heightened in non-treated ‘*Ca*. L. asiaticus’-infected leaves (91.1 ± 13.6 ng g^−1^ FW) compared to non-treated healthy control (62.8 ± 12.5 ng g^−1^ FW), however, its levels were also increased upon GABA application in treated ‘*Ca*. L. asiaticus’-infected plants (180.7 ± 48.5 ng g^−1^ FW), but not in healthy ones (83.2 ± 14.0 ng g^−1^ FW) (Table 1).

### 2.2. GABA Accumulation Alters the Endogenous Levels of Other Amines in Both Healthy and ‘Ca. L. asiaticus’-Infected Plants

After MCF derivatization, four PhA (*ρ*-aminobenzoic acid, tyramine, synephrine, and octopamine) and three NPAA (GABA, pyroglutamic acid, and *O*-acetyl serine) were detected in all treatments (Figure 1). Although ‘*Ca*. L. asiaticus’ infection did not affect the content of *ρ*-aminobenzoic acid (Figure 1A) and tyramine (Figure 1B), it significantly enhanced the accumulation of synephrine (Figure 1C), octopamine (Figure 1D), GABA (Figure 1E), and pyroglutamic acid (Figure 1F). It is worth mentioning that GABA application did not affect the endogenous levels of any of these seven metabolites in GABA-treated healthy leaves compared with the non-treated control. However, exogenous GABA application significantly increased the endogenous levels of tyramine (Figure 1B), octopamine (Figure 1D), and GABA (Figure 1E) in ‘*Ca*. L. asiaticus’-infected plants compared with non-treated ones. Likewise, in ‘*Ca*. L. asiaticus’-infected leaves, endogenous GABA content was increased upon exogenous GABA application (*p* = 0.0015) compared with non-treated infected plants (Figure 1E). On the other hand, ‘*Ca*. L. asiaticus’ infection significantly reduced the levels of pyroglutamic acid in non-treated leaves compared with non-treated control (*p* = 0.0009; Figure 1F). Nevertheless, neither GABA application nor ‘*Ca*. L. asiaticus’ infection affected the *O*-acetyl serine content in both healthy and infected citrus plants (Figure 1G).

### 2.3. Exogenous GABA Application Alters the Organic Acids and TCA-Associated Compounds in Healthy and ‘Ca. L. asiaticus’-Infected Plants

Using a non-targeted GC-MS-based method, 10 organic acids were detected in citrus leaves (Figure 2). Out of these ten compounds, five metabolites were associated with the TCA cycle including citric acid (Figure 2A), *α*-ketoglutaric acid (Figure 2B), succinic acid (Figure 2C), fumaric acid (Figure 2D), and malic acid (Figure 2E). ‘*Ca*. L. asiaticus’ infection significantly enhanced the accumulation of citric acid, succinic acid, and fumaric acid, but markedly reduced the levels of *α*-ketoglutaric acid and malic acid compared with the non-treated healthy plants. It is worth mentioning that exogenous GABA application induced the accumulation of both citric acid and succinic acid and significantly reduced the content of malic acid, but did not affect either *α*-ketoglutaric acid or fumaric acid in GABA-treated healthy plants compared with non-treated control. Interestingly, the earliest TCA-associated compound, citric acid, was significantly increased in ‘*Ca*. L. asiaticus’-infected leaves (13.9 ± 2.2 μg g^−1^ FW) compared to non-treated healthy control (3.3 ± 0.7 μg g^−1^ FW); however, its levels were also increased upon GABA application in both healthy (8.3 ± 1.9 μg g^−1^ FW) and ‘*Ca*. L. asiaticus’-infected plants (21.0 ± 6.1 μg g^−1^ FW) compared with non-treated control (Figure 2A). On the other hand, the most abundant OA, malic acid, was significantly reduced upon ‘*Ca*. L. asiaticus’-infection (29.8 ± 0.5 μg g^−1^ FW) compared with non-treated control (33.5 ± 1.4 μg g^−1^ FW); nevertheless, root drench application of 10 mM GABA negatively affected its levels in both treated healthy (29.5 ± 0.7 μg g^−1^ FW) and infected (28.7 ± 0.7 μg g^−1^ FW) plants compared with the non-treated control (Figure 2E).

Moreover, three metabolites associated with the SA biosynthesis pathway were also detected within the OA profile in both healthy and ‘*Ca*. L. asiaticus’-infected citrus plants, including *t*-cinnamic acid (*t*CA; Figure 2F), benzoic acid (BA; Figure 2G), and salicylic acid (SA; Figure 2H). Although ‘*Ca*. L. asiaticus’ infection resulted in a higher accumulation of *t*CA (1.3 ± 0.4 μg g^−1^ FW), BA (4.9 ± 0.6 μg g^−1^ FW), and SA (2.8 ± 0.7 μg g^−1^ FW) compared with non-treated control (0.7 ± 0.1, 2.9 ± 0.6, and 0.9 ± 0.2 μg g^−1^ FW, respectively), GABA application stimulated the accumulation of only SA in both healthy (2.1 ± 0.2 μg g^−1^ FW) and ‘*Ca*. L. asiaticus’-infected plants (4.7 ± 0.7 μg g^−1^ FW). Remarkably, GABA-treated infected plants had the highest *t*CA, BA, and SA content throughout the experiment (2.1, 2.9, and 5.2 fold higher than the non-treated healthy plants, respectively).

Additionally, two more OA were detected in citrus leaf extract including quinic acid (Figure 2I) and ferulic acid (Figure 2J). Although quinic acid was significantly reduced in ‘*Ca*. L. asiaticus’-infected plants compared to non-treated control (*p* = 0.0003), exogenous GABA supplementation significantly enhanced its accumulation in infected plants (up to 3.2-folds), but not healthy ones. On the other hand, ferulic acid was reduced in ‘*Ca*. L. asiaticus’-infected plants (2.1 ± 0.5 μg g^−1^ FW) compared with non-treated control (3.2 ± 0.6 μg g^−1^ FW); however, GABA application significantly enhanced its endogenous levels in infected plants (5.3 ± 0.5 μg g^−1^ FW), but not in healthy (4.3 ± 0.8 μg g^−1^ FW) ones.

### 2.4. GABA Supplementation Induced the Accumulation of Fatty Acids in Healthy and ‘Ca. L. asiaticus’-Infected Plants

Five fatty acids were detected in citrus leaves methanolic extract using GC-MS running in the full scan mode, including palmitic (Figure 3A), stearic (Figure 3B), oleic (Figure 3C), linoleic (Figure 3D), and α-linolenic (Figure 3E) acids. The levels of detected FA were significantly increased in ‘*Ca*. L. asiaticus’-infected leaves compared to non-treated healthy control, except stearic acid which was not altered significantly. Likewise, root drench application of 10 mM GABA noticeably enhanced the FA profile of healthy citrus trees, except for stearic acid. It is worth mentioning that α-linolenic acid (C18:3; the precursor of jasmonic acid (JA)) was significantly heightened in ‘*Ca*. L. asiaticus’-infected plants (3.04 ± 0.75 μg g^−1^ FW) compared to the non-treated control (0.61 ± 0.20 μg g^−1^ FW). Nevertheless, GABA supplementation stimulated the accumulation of α-linolenic acid in treated healthy plants (1.45 ± 0.37 ng g^−1^ FW) compared with non-treated healthy control (*p* = 0.0007), but not ‘*Ca*. L. asiaticus’-infected ones (3.63 ± 0.28 ng g^−1^ FW) when compared with non-treated infected ones (*p* = 0.0985; Figure 3E).

### 2.5. Principal Component Analysis Reveals Differences in Detected Metabolites between GABA-Treated and Non-Treated Healthy and ‘Ca. L. asiaticus’-Infected Citrus Plants

To better understand the impact of GABA application on the whole metabolic profile of both healthy and ‘*Ca*. L. asiaticus’-infected citrus plants, principal component analysis (PCA), and two-way hierarchical cluster analysis (HCA) were carried out (Figure 4). Briefly, the PCA-associated scatter plot showed a clear separation between healthy and ‘*Ca*. L. asiaticus’-infected plants. Moreover, the PCA scatter plot showed clear discrimination between GABA-treated and non-treated plants within the same treatment (Figure 4A). PC1 and PC2 were responsible for approximately 88% of the variation (75.23 and 12.98%, respectively). Additionally, the PCA-associated loading plot showed that while eight metabolites were positively correlated with healthy plants (either treated or not), 17 metabolites were associated with GABA-treated, ‘*Ca*. L. asiaticus’-infected leaves, while the rest (16 compounds) were associated with non-treated ‘*Ca*. L. asiaticus’-infected plants.

Briefly, 2-ketoglutaric acid, malic acid, _L_-pyroglutamic acid, _L_-glutamic acid, and _L_-methionine were associated with the non-treated healthy control, whereas only three metabolites (_L_-asparagine, _L_-glutamine, and _L_-cysteine) were correlated positively with the GABA-treated healthy control (Figure 4B).

In harmony with PCA results, HCA and its associated heatmap (Figure 4C) revealed the differences among the four studied treatments. Briefly, HCA-associated dendrogram among treatments showed that GABA-treated healthy control was closer to non-treated infected plants. Additionally, the HCA-associated dendrogram among metabolites showed that all examined metabolites were clustered into three distinct clusters. “Cluster I” consists of 33 compounds which were significantly higher in GABA-treated infected plants than in other treatments. It is worth mentioning that “Cluster I” included the major phytohormones’ precursors including _L_-phenylalanine, _L_-tryptophan, and *α*-linolenic acid (C18:3), the precursors of SA, auxins, and JA, respectively. Moreover, the total HCA-dendrogram showed that “Cluster II” consists of only five metabolites including 2-ketoglutaric acid, malic acid, _L_-pyroglutamic acid, _L_-glutamic acid, and _L_-methionine, which were associated with the non-treated healthy control, whereas “Cluster III” consists of only three metabolites including _L_-asparagine, _L_-glutamine, and _L_-cysteine (Figure 4C).

### 2.6. Enhanced GABA Levels Alleviate Oxidative Stress in Both Healthy and ‘Ca. L. asiaticus’-Infected Citrus Plants

‘*Ca*. L. asiaticus’ might generate oxidative stress in infected citrus plants and GABA accumulation may play a key role in alleviating this harmful effect. To test this hypothesis, NBT- and DAB-based staining was used to analyze the accumulation of O_2_^•−^ and H_2_O_2_, respectively (Figure 5). Briefly, in situ histochemical localization of O_2_^•−^ using NBT-based staining showed that non-treated ‘*Ca*. L. asiaticus’-infected leaves accumulated more O_2_^•−^ as indicated by darker blue color in the midrib and lamina areas (Figure 5A) and higher integrated optical density (0.068 ± 0.034 IOD) than the non-treated healthy control (0.010 ± 0.009 IOD). However, GABA supplementation significantly diminished the O_2_^•−^ accumulation in both healthy (0.012 ± 0.008 IOD) and infected plants (0.038 ± 0.009 IOD) (Figure 5B).

Likewise, in situ histochemical localization of H_2_O_2_ using DAB-based staining showed that the non-treated ‘*Ca*. L. asiaticus’-infected leaves accumulated darker brown colored spots (an indicator of H_2_O_2_) than other treatments (Figure 5C) as indicated by integrated optical density (0.523 ± 0.098 IOD) compared with the non-treated control (0.205 ± 0.107 IOD) which indicates higher levels of H_2_O_2_ were produced upon ‘*Ca*. L. asiaticus’ infection (Figure 5D). Nevertheless, exogenous GABA application markedly reduced the accumulation of H_2_O_2_ within healthy (0.175 ± 0.066 IOD) and ‘*Ca*. L. asiaticus’-infected leaves (0.265 ± 0.059 IOD) (Figure 5D).

Moreover, the H_2_O_2_ accumulation was confirmed using a colorimetric method and quantified using the H_2_O_2_ standard curve (Figure 5E). Colorimetric spectrophotometric determination of H_2_O_2_ showed that non-treated ‘*Ca*. L. asiaticus’-infected leaves accumulated higher levels of H_2_O_2_ (557.56 ± 121.63 nmol g^−1^ FW) compared with non-treated healthy ones (299.98 ± 97.57 nmol g^−1^ FW). However, GABA application significantly reduced the accumulation of H_2_O_2_ in both healthy (215.34 ± 51.34 nmol g^−1^ FW) and infected leaves (258.08 ± 77.43 nmol g^−1^ FW) (Figure 5F).

Furthermore, ROS accumulation was fluorescently localized using H_2_DCFDA-based staining (Figure 6). ROS is indicated as a green fluorescence mainly distributed in the background and/or around the leaf edges. In non-treated leaves, ‘*Ca*. L. asiaticus’-infection resulted in a significant increase in ROS as indicated by stronger green fluorescence compared to the non-treated healthy control; however, GABA application significantly reduced the ROS accumulation in GABA-treated healthy and infected leaves (Figure 6A). It is worth mentioning that ROS accumulation was more pronounced in the laminar abscission zone (area between the leaflet blade and petiole wing), particularly when ‘*Ca*. L. asiaticus’ was present (Figure 6B). Likewise, ROS accumulation was more noticeable in the HLB-symptomatic areas of leaf lamina with higher levels of green fluorescence than in non-symptomatic control leaves (Figure 6C,D). Nevertheless, our fluorescence microscopy findings suggest that exogenous GABA application alleviates the oxidative stress in ‘*Ca*. L. asiaticus’-infected citrus leaves (Figure 6). The 3D surface plotting of relative fluorescence intensity supports these findings (Figure 6E).

### 2.7. GABA Accumulation Augments the Transcript Levels of Antioxidant-Related Genes Both Healthy and ‘Ca. L. asiaticus’-Infected Plants

To better understand how GABA alleviate the oxidative stress in citrus leaves, the impact of exogenous GABA application on the transcript levels of 11 antioxidant-related genes (Figure 7) in healthy and ‘*Ca*. L. asiaticus’-infected citrus leaves were investigated. These enzymes included ascorbate peroxidase, cytosolic isoform X1 (*CsAPX*), phospholipid hydroperoxide glutathione peroxidase (*CsGPX*), cationic peroxidase 1-like (*CscPOX*), lignin-forming anionic peroxidase-like (*CsliPOX*), peroxidase A2-like (*CsPOX-A2*), peroxidase 3 (*CsPOX3*), superoxide dismutase [Cu-Zn], chloroplastic (*CsSOD-Cu/Zn*), superoxide dismutase [Mn], mitochondrial (*CsSOD-Mn*), superoxide dismutase [Fe] 2, chloroplastic-like (*CsSOD-Fe*), superoxide dismutase [Fe] 3, chloroplastic isoform X1 (*CsSOD-Fe3*), and catalase-like isoform X1 (*CsCAT*). Briefly, ‘*Ca*. L. asiaticus’ infection slightly increased the transcript levels of *CsAPX* (*p* = 0.0001; Figure 7A), *CsGPX* (*p* = 0.0001; Figure 7B), *CscPOX* (*p* = 0.0001; Figure 7C), *CsliPOX* (*p* = 0.0001; Figure 7D), *CsPOX-A2* (*p* = 0.0001; Figure 7E), *CsPOX3* (*p* = 0.0001; Figure 7F), *CsSOD-Cu/Zn* (*p* = 0.0001; Figure 7G), *CsSOD-Mn* (*p* = 0.0001; Figure 7H), *CsSOD-Fe* (*p* = 0.0001; Figure 7I), *CsSOD-Fe3* (*p* = 0.0001; Figure 7J), and *CsCAT*(*p* = 0.0001; Figure 7K). However, all studied antioxidant-related genes were significantly upregulated (up to 6.9 fold) in GABA-treated leaves with a greater effect in the ‘*Ca*. L. asiaticus’-infected plants (Figure 7).

### 2.8. Exogenous GABA Application Alters the Transcript Levels of Genes Implicated in Multiple Metabolic Pathways

To better understand the impact of GABA accumulation on different metabolic pathways in healthy and ‘*Ca*. L. asiaticus’-infected plants, the transcript levels of 67 genes involved in GABA shunt (18 genes; Figure 8A), AA-related pathways (12 genes; Figure 8B), and phytohormones biosynthesis pathways (37 genes; Figure 8C), including SA (nine genes), JA (nine genes), auxins (ten genes), abscisic acid (ABA; six genes), and ethylene (ET; three genes). Generally, the transcript levels of all investigated genes were increased after GABA supplementation with a greater effect on ‘*Ca*. L. asiaticus’-infected plants (Figure 8). In comparison with the non-treated healthy control, GABA-treated infected plants had the highest expression levels of the three *gab* genes among all GABA shunt-related genes. These genes included GABA permease (*CsgabP*; aka amino acid permease BAT1-like, approximately 7.9 folds), GABA transaminase 3 (*CsgabT*, up to 8.9-folds), and GABA dehydrogenase (*CsgabD*; also known as succinate-semialdehyde dehydrogenase (SSADH), up to 8.7 folds) (Figure 8A).

In terms of AA-related genes, tyrosine decarboxylase 1 (*CsTDC1*), arginine decarboxylase-like (*CsADC*), and serine acetyltransferase 5 (*CsSAT5*) were the highest expressed genes in GABA-treated infected plants (around 5 folds) compared to non-treated healthy plants. Moreover, proline dehydrogenase 1 (*CsProDH*) and δ-1-pyrroline-5-carboxylate dehydrogenase 12A1(*CsP5CDH*), genes implicated in glutamine and proline metabolism, were also expressed at higher levels in GABA-treated infected plants compared to non-treated healthy ones (4.2 and 4.3 folds, respectively) (Figure 8B).

Moreover, the transcript levels of the nine studied SA biosynthesis genes were boosted after GABA application with a greater effect in the ‘*Ca*. L. asiaticus’-infected plants (Figure 8C). Chorismate mutase (*CsCM*), arogenate dehydratase/prephenate dehydratase 1, chloroplastic (*CsADT*), aspartate aminotransferase, cytoplasmic-like (*CsAST*), isochorismate synthase (*CsICS*), and 3-ketoacyl-CoA thiolase 2 (*CsKAT2*) had the highest expression levels in GABA-treated infected plants compared to non-treated healthy ones (5.2, 5.6, 5.2, 5.7, and 6.5 folds, respectively). On the other hand, although GABA application upregulated all JA biosynthesis-related genes, allene oxide synthase (*CsAOS*), allene oxide cyclase (*CsAOC*), acetate/butyrate CoA ligase AAE7 (*CsAAE7*), and 12-oxophytodienoate reductase 3 (*CsOPR3*) had the highest transcript levels in GABA-treated ‘*Ca*. L. asiaticus’-infected plants compared with non-treated healthy control (6.0, 6.6, 5.6, and 6.3 folds, respectively) (Figure 8C).

Likewise, the gene expressions of all auxins biosynthesis-related genes were upregulated in GABA-treated ‘*Ca*. L. asiaticus’-infected plants compared to non-treated control (up to 6.5 folds). It is worth mentioning that anthranilate synthase beta subunit 2 (*CsASB*), tryptophan synthase alpha chain, chloroplastic-like (*CsTSA*), and tryptophan aminotransferase-related protein 4-like (*CsTAA4*) were the highest expressed genes in ‘*Ca*. L. asiaticus’-infected plants upon GABA supplementation (5.6, 6.5, and 5.4 folds, respectively). Furthermore, GABA application upregulated all studied ABA biosynthesis-related genes in both healthy and ‘*Ca*. L. asiaticus’-infected plants. These genes included zeaxanthin epoxidase (*CsZEP*; up to 6.1 folds), violaxanthin de-epoxidase (*CsVDE*; up to 5.2 folds), neoxanthin synthase (*CsNSY*; up to 5.9 folds), 9-*cis*-epoxycarotenoid dioxygenase 3 (*CsNCED*; up to 5.6 folds), short-chain alcohol dehydrogenase (*CsABA2*; up to 4.9 folds), and abscisic aldehyde oxidase (*CsAAO3*; up to 6.2 folds) compared with non-treated healthy control (Figure 8C). Likewise, three ET biosynthesis-related genes were upregulated in both healthy and infected citrus plants upon GABA application in comparison with the non-treated healthy control. These genes included 1-aminocyclopropane-1-carboxylate (ACC) synthase-like (*CsACS*), ACC oxidase (*CsACO*), and *S*-adenosylmethionine decarboxylase (*CsSAMDC*) (Figure 8C). Collectively, the gene expression findings support our findings from the GC-MS work.

## 3. Discussion

Land plants employ a complex defense system to counteract the infection with phytopathogens and other biotic stressors [59,60,61]. Reprogramming of plant metabolism is a key strategy in this complex immune response, mainly to ensure the required energy for the biosynthesis of defense-related metabolites [62,63,64]. Our previous studies showed that ‘*Ca*. L. asiaticus’ infection and/or *D. citri* infestation altered several metabolic pathways in their host including VOCs [65], carboxylic compounds [17], leaf pigments [18], phytohormones [19], phytomelatonin [21], polyamines, and NPAA, particularly GABA [20,66,67].

These metabolic responses might be a result of host cellular functions for defense reactions, or they might be manipulated by ‘*Ca*. L. asiaticus’ and/or its vector for their benefit to fulfill their nutritional needs, or to induce symptoms development. For instance, some biotrophic and hemibiotrophic phytopathogens benefit from elevated GABA content in the apoplast and use it as a nitrogen (N) source [68,69]. GABA was reported as the main N source for the fungal pathogen *Cladosporium fulvum* and the bacterial pathogen *Pseudomonas syringae* pv. *Tomato* DC3000 (*Pst*- DC3000) during their infection in tomato [68,69]. It is worth mentioning that the genome possesses three GABA transaminase genes (*gabT*) [45]. Deletion mutants of *gabT* in *Pst*- DC3000 were incapable of using GABA as a sole nitrogen source and negatively affected its growth and virulence [45].

GABA was reported previously to be accumulated upon infection with numerous phytopathogenic bacteria such as *Pseudomonas syringae* on *Arabidopsis thaliana* [45] and *Phaseolus vulgaris* [46], *Agrobacterium tumefaciens* on *A. thaliana* [47], *Xylella fastidiosa* on *Vitis vinifera* [48], *Ralstonia solanacearum* on *Solanum lycopersicon* [49], and ‘*Ca*. L. asiaticus’ on *C. sinensis* [17,20,54]. The induction of endogenous GABA content in infected parts might be a part of host defensive responses; however, the mode of action for GABA seems to be pathogen dependent. For instance, while the GABAergic effects on the development of insect larvae and root-knot nematodes were associated with interrupting the neuromuscular junctions or via direct jasmonate-independent defense response [43,44,70], the potential roles GABA against bacterial phytopathogens include down-regulation of quorum sensing (QS) [71], modulation of host hypersensitive response (HR) [72], and/or modification of redox status in the host plant [73].

Quorum sensing (QS) is a process of cell-to-cell communication in bacteria that coordinates their behavior and regulates their gene expression based on the population density [74,75]. The LuxR/LuxI-type QS system is one of the most extensively studied QS systems in Gram-negative bacteria [76]. The LuxI protein is an autoinducer synthase that catalyzes the formation of a specific acyl-homoserine lactone (AHL) molecule [76]. The AHL molecule diffuses through the cell membrane at high cell density and binds to the LuxR protein, which is a transcriptional regulator [76]. The LuxR-AHL complex then activates the transcription of its target genes [76]. QS was proposed previously to be crucial for the growth and multiplication of ‘*Ca*. L. asiaticus’ [77]. It is worth mentioning that the genome of ‘*Ca*. L. asiaticus’ possesses the LuxR protein, one of a two-component cell-to-cell communication system [78]. For instance, the transcription levels of two putative LuxR family transcriptional regulators were significantly increased in the host plant [77]. However, the genome lacks the second component, LuxI, that produces acyl-homoserine lactones (AHLs), suggesting that ‘*Ca*. L. asiaticus’ has a solo LuxR system [78]. The absence of LuxI suggests that ‘*Ca*. L. asiaticus’ may use alternative signaling molecules to regulate gene expression [78]. However, the exact mechanism by which ‘*Ca*. L. asiaticus’ regulates gene expression in the absence of AHLs is not yet fully understood. It has been suggested that other signaling molecules such as GABA may be involved in regulating virulence genes and biofilm formation in ‘*Ca*. L. asiaticus’ [75]. We hypothesize that induced GABA levels in infected plants might be for the benefit of the pathogen itself or it is presumably part of citrus’s defensive response role against ‘*Ca*. L. asiaticus’ to interfere with its QS signaling system. Nevertheless, because ‘*Ca*. L. asiaticus’ is not cultured yet, neither the exact mechanism by which GABA regulates virulence genes and biofilm formation in ‘*Ca*. L. asiaticus’, nor the molecular mechanism(s) behind “how does GABA interfere with the QS signaling system?” is yet fully understood and requires further research.

Moreover, we hypothesize GABAergic effects in ‘*Ca*. L. asiaticus’-infected citrus plants are linked with other metabolic pathways. In the current study, non-targeted metabolomic analysis of citrus leaves revealed GABA accumulation, either due to ‘*Ca*. L. asiaticus’ infection or exogenous GABA supplementation, altered multiple metabolic pathways including PAA, NPAA, PhA, OA, FA, and phytohormones, as well as several biosynthetic genes involved in these pathways (Figure 9). Partially, similar findings were reported previously upon the infection with ‘*Ca*. L. asiaticus’ for others [17,19,20,54,79,80,81] and ‘*Ca.* L. solanacearum’, the pathogen of Zebra chip disease of potato [82,83,84]. GABA accumulation is also associated with the regulation of these metabolic pathways.

For instance, the glutamine/glutamate metabolism pathway delivers the carbon backbone and an amino group to produce several defense-related metabolites such as proline and GABA [63,73]. Our findings showed that _L_-glutamic acid did not affect significantly due to ‘*Ca*. L. asiaticus’ infection or exogenous GABA application, _L_- glutamine was reduced in infected citrus plants, which resulted in the reduction of the total glutamate pool. This might be due to the conversion of glutamate to GABA or proline (Figure 9). _L_-Proline was the most abundant AA in citrus leaves, and it accumulated to higher levels in ‘*Ca*. L. asiaticus’-infected and GABA-treated plants. Proline is synthesized from glutamate [80,85] using the activity of *P5CDH* and *ProDH* which both were upregulated upon ‘*Ca*. L. asiaticus’ infection and exogenous GABA application. Additionally, endogenous GABA levels and the transcript level of the key GABA-biosynthesis genes glutamate decarboxylase-like (*GAD*) and glutamate decarboxylase 5-like (*GAD5*) were significantly increased upon ‘*Ca*. L. asiaticus’ infection and exogenous GABA application. Both proline and GABA and proline biosynthesis (*P5CDH* and *ProDH*) protect the ‘*Ca*. L. asiaticus’-infected plants from ROS [84,86]. Collectively, these findings support our hypothesis that glutamate reduction partially might be due to its conversion to proline and GABA [85,87] to maintain the redox status in infected plants.

Moreover, we suggest that GABA accumulation plays a key role in the regulation of multiple phytohormone biosynthesis pathways including auxins, SA, JA, ET, and ABA. Although the role of GABA in plants is well-documented, and it was reported as a modulator of phytohormones in several plant species under abiotic stress [88,89,90], and in non-stressed healthy citrus plants, [53] the question of “How do plant hormones interact with GABA?” is still an open question and required further study [91]. Herein, we answered this question in part. Briefly, our findings showed that induced GABA levels, due to ‘*Ca*. L. asiaticus’ infection or exogenous GABA application, were associated with the induction of _L-_tryptophan, _L_-phenylalanine, and α-linolenic acid, the main precursors of auxins, SA, and JA, respectively. Moreover, GABA supplementation upregulated most, if not all, of phytohormones biosynthesis genes in both healthy and ‘*Ca.* L. asiaticus’-infected leaves. Previously, we showed that while SA was involved in citrus defense response against ‘*Ca*. L. asiaticus’, JA was associated with citrus response to its insect vector, *D. citri* [19]. Furthermore, our previous studies proved that exogenous GABA application positively regulates endogenous levels of various phytohormones (auxins, SA, JA, and ABA) in healthy citrus plants [53]. Taken together, these findings suggest that GABA accumulation, whatever the reason, stimulates the biosynthesis pathways of multiple phytohormones at its earliest steps (precursors) and induces the expression levels of major phytohormones-associated biosynthetic genes.

Unfortunately, the potential roles of GABA accumulation in hormone metabolism are poorly reported from biotic-stressed citrus plants. However, very few reports are available about its role in the biosynthesis and metabolism of the gaseous phytohormone ethylene (ET) [27,88]. The volatile hormone, ET, is synthesized from the AA methionine via a simple two-step biosynthesis pathway using 1-aminocyclopropane-1-carboxylic acid (ACC) synthase (*ACS*) and ACC oxidase (*ACO*) [92]. Although we cannot detect ET using our method, gene expression analysis showed that both *ACS* and *ACO* were upregulated upon ‘*Ca*. L. asiaticus’ infection and exogenous GABA application indicate the induction of the ET biosynthesis pathway. Moreover, *S*-adenosyl-methionine decarboxylase (*SAMDC*; aka *AMD1*) is a key enzyme that connects both ET and PA biosynthesis pathways [67]. Both ET and PAs play diverse roles in citrus defense responses to biotic and abiotic stress [66]. However, the molecular and biochemical mechanisms behind ET-*SAMDC*-PA’s defensive role(s) remain poorly understood. *SAMDC* from navel orange (*C. sinensis*; GenBank Accession No. FJ496345) was cloned and biochemically identified as a rate-limiting enzyme for PA biosynthesis [93]. Our findings showed that *SAMDC* was upregulated upon GABA application in both healthy and ‘*Ca*. L. asiaticus’-infected plants. Together, these findings suggest GABA regulates both pathways (ET and PAs) to refine citrus defensive responses against HLB; however, their exact roles are still unclear and require further study.

Moreover, GABA accumulation might boost the plant’s defensive responses against phytopathogenic microbes via augmenting TCA cycle activity [20,37]. In the current study, our findings showed that GABA accumulation altered the endogenous levels of several TCA-associated OA including citrate, *α*-ketoglutarate, succinate, fumarate, and malate. Previous studies showed that citrate was accumulated in ‘*Ca*. L. asiaticus’-infected leaves and fruits [79,94], as well as succinate was accumulated in infected fruits [79]. Conceivably, the accumulation of TCA-associated compounds is due to the catabolism of some PAA [95]. Briefly, glucogenic AA are catabolized into pyruvate, α- ketoglutarate, succinyl CoA, fumarate, or oxaloacetate, whereas ketogenic AA are catabolized into acetyl-CoA or acetoacetate [96]. In the current study, even though citrate, succinate, fumarate, and malate were significantly increased upon GABA accumulation due to ‘*Ca*. L. asiaticus’ infection and GABA application, *α*-ketoglutarate was notably decreased. Low α-ketoglutarate content might be due to the hastening of its catabolism into glutamate (_L_-glutamine and _L_-glutamic acid) using the enzymatic activity of glutamate dehydrogenase (*GDH*), glutamate synthase (*GS*), γ-Glutamylcysteine synthetase (γ-*GCS*), and glutamate 5-kinase (*G5K*) [97,98]. All these enzymes were mostly expressed at higher levels upon the ‘*Ca*. L. asiaticus’ infection and GABA application. Interestingly, the endogenous levels of _L_-glutamine, _L_-glutamic acid, and pyroglutamic acid were also reduced upon ‘*Ca*. L. asiaticus’ infection and GABA application result in an accumulation of cytosolic GABA.

In citrus plants, GABA-shunt is functionally linked with the TCA cycle [20] via three *gab* genes that enable the non-cyclic flux from *α*-ketoglutarate to succinate in ‘*Ca*. L. asiaticus’-infected citrus plants [54]. In parallel with this suggestion, citrate catabolism was found to be connected with GABA shunt in citrus fruits [99,100]. Briefly, citrate is catalyzed into α-ketoglutarate then glutamate which is decarboxylated into GABA in the cytosol, via isocitrate dehydrogenase (*IDH*), glutamate dehydrogenase (*GDH*), glutamate synthase (*GS*), glutamate decarboxylase (*GAD*), and the activities of many other enzymes (Figure 9) [101,102]. Subsequently, cytosolic GABA is transported from the cytosol into the mitochondria using mitochondrial GABA permease (*gabP*) [20,54,103]. Once GABA is inside the mitochondria, it is catabolized to succinic semialdehyde, then to succinate using GABA aminotransferase (*gabT*) and GABA dehydrogenase (*gabD*), respectively [54]. Recently, we showed that the citrus genome harbors these three *gab* genes (*gabP*, *gabT*, and *gabD*). Our findings showed that all *gab* genes were upregulated after ‘*Ca*. L. asiaticus’ infection and after the exogenous GABA application. These findings indicate that GABA might be catabolized into succinate directly under high GABA conditions rather than an intact TCA cycle.

In addition to metabolic reprogramming, GABA plays a key role in the up-regulation of antioxidant defense systems to maintain the redox status within stressed plants [55,104,105]. Under stressful conditions, ROS, such as superoxide radicals, singlet oxygen, H_2_O_2_, and O_2_^•−^, are produced at the plasma membrane and accumulate in different organelles including the chloroplast, mitochondria, and peroxisomes as a part of host defense responses [106]. However, overproduction and accumulation of ROS are cytotoxic, can cause cellular damage, and lead to programmed cell death [106,107], and therefore it requires tight control. In the current study, in situ histochemical localization and visualization of H_2_O_2_ and O_2_^•−^ showed that infection with ‘*Ca*. L. asiaticus’ induced the accumulation of both ROS and generated strong oxidative stress in infected leaves. Accumulation of H_2_O_2_ in ‘*Ca*. L. asiaticus’-infected citrus might be due to the alteration of polyamine metabolism and their catabolic genes including diamine oxidase (*DAO*) and polyamine oxidase (*PAO*) [56]. The overabundance of H_2_O_2_ eventually becomes toxic to the leaf tissue and leads to the development of the HLB characteristic blotchy mottle symptom (well-reviewed by [5]). It is worth mentioning that ‘*Ca*. L. asiaticus’ possesses its own peroxidase that enables it to survive the toxic effects of H_2_O_2_ accumulation [56,108]. Nevertheless, the antioxidant defense system of citrus is presumably not sufficient to lessen the ROS accumulation, particularly H_2_O_2_, which may eventually become toxic to the leaf tissue [5,56]. However, our findings showed that GABA accumulation due to ‘*Ca*. L. asiaticus’ infection or exogenous GABA supplementation remarkably reduced the accumulation of H_2_O_2_ and O_2_^•−^ in infected leaves.

These findings suggest that GABA possibly stimulates the activation of a multilayered antioxidative system to neutralize the deleterious effect of ROS and to maintain redox status within infected leaves. This complex antioxidant defense system comprises two major components, enzymatic and non-enzymatic machinery. The enzymatic machinery provides the first defense line, whereas non-enzymatic machinery works as the second line of defense against ROS [106,109,110,111]. The enzymatic antioxidant machinery comprises multiple antioxidant-related enzymes including several peroxidases (POX), superoxide dismutases (SOD), and catalase (CAT) [110,111,112]. Antioxidant-related enzymes directly scavenge ROS, particularly H_2_O_2_ and O_2_*^•^*^−^, to reduce their reactivity [106,110]. In the current study, the transcription of six peroxidases (*CsAPX*, *CsGPX*, *CscPOX*, *CsliPOX*, *CsPOX-A2*, and *CsPOX3*), four superoxide dismutases (*CsSOD-Cu/Zn*, *CsSOD-Mn*, *CsSOD-Fe*, and *CsSOD-Fe3*), and catalase-like isoform X1 (*CsCAT*) were increased in both healthy and ‘*Ca.* L. asiaticus’-infected leaves after GABA application. Together, these findings indicate that GABA triggers the enzymatic antioxidant defense machinery to help citrus plants manage the detrimental oxidative stress caused by ‘*Ca*. L. asiaticus’.

Furthermore, the non-enzymatic antioxidant defense machinery relay on hydrophilic metabolites such as phenolics flavonoids and carotenoids [109,110]. Although our previous study showed that infection with ‘*Ca*. L. asiaticus’ significantly reduced the levels of most, if not all, carotenoids in infected leaves [18], our findings from the current study showed that GABA application significantly increased the endogenous levels of several phenolic metabolites. For example, GABA application significantly increased the endogenous levels of four phenolic amines (*ρ*-aminobenzoic acid, tyramine, synephrine, and octopamine) in both healthy and ‘*Ca*. L. asiaticus’-infected plants compared with non-treated control. Likewise, although infection with ‘*Ca*. L. asiaticus’ reduced the endogenous content of two major phenolic acids (quinic acid and ferulic acid), and exogenous GABA application significantly induced the accumulation of both phenolic acids in ‘*Ca*. L. asiaticus’-infected leaves. We hypothesize that GABA is involved in the activation of the non-enzymatic antioxidant defense machinery via the accumulation of phenolic compounds (amines and acids) that may act as a second line of defense against ROS [109,110,111].

## 4. Materials and Methods

### 4.1. Plant Materials and Growth Conditions

All experiments were carried out at the Citrus Research and Education Center, the University of Florida (CREC-UF), Lake Alfred, Florida, using the HLB-susceptible cultivar ‘Valencia’ sweet orange (*Citrus sinensis*) as an experimental plant. All plant materials were produced by grafting and were approximately 100 ± 5 cm in height and around 18 months old at sampling time. ‘Valencia’ trees were kept under greenhouse conditions at 27 ± 2 °C, with 70 ± 5% relative humidity, and an 8:16 dark/light cycle. Citrus trees were irrigated twice weekly and fertilized monthly with a water-soluble 20:10:20 NPK fertilizer (Allentown, PA, USA). To obtain the ‘*Ca*. L. asiaticus’-infected trees, 12-month-old healthy plants were inoculated via bud grafting using ‘*Ca*. L. asiaticus’-positive materials in March 2020, and maintained under the same conditions described above. Upon the development of the initial HLB symptoms, the presence of ‘*Ca*. L. asiaticus’ was confirmed at six months post inoculation (mpi) using PCR as described by [113]. Healthy plants were also grafted using ‘*Ca*. L. asiaticus’-negative materials to avoid the grafting effect.

### 4.2. Treatment of Citrus Plants with Exogenous GABA and Leaf Sampling

To investigate the impact of GABA accumulation on the homeostasis of several primary and secondary metabolic pathways, each tree of ‘Valencia’ sweet orange (healthy versus ‘*Ca*. L. asiaticus’-infected) was supplemented via soil drench with 300 mL of aqueous GABA solution (Sigma-Aldrich, St. Louis, MO, USA) at a final concentration of 10 mM based on our previous studies [53,54,114]. Non-treated controls were treated with 300 mL of distilled water. The solution that exceeded the soil field capacity, and gravitationally drained from the pot bottom, was collected and reapplied to the soil. All plant materials (healthy vs. infected and GABA-treated vs. non-treated) were treated in October 2020 and maintained under the greenhouse conditions described above. Our previous studies showed that the positive effects of exogenous GABA application reached their highest peak at 7 days post treatment (dpt) [53,114]. Therefore, all plant materials for all below analyses were collected at 7 dpt.

For sampling, three leaves were collected from each biological sample, from different positions and different ages: juvenile leaves from the top, intermediate-aged leaves (fully expanded, but not hardened) from the middle, and mature leaves (deep green and hardened) from the lower part of the plant. The collected leaves were homogenized, mixed together, and immediately kept at −80 °C until further analysis. For the in situ histochemical localization of hydrogen peroxide (H_2_O_2_) and superoxide anion (O_2_^•−^) assays, 20 leaf discs (10 from the lamina and 10 from the midrib, approximately 1 cm^2^ each) were used for each treatment.

### 4.3. Non-Targeted Metabolomics Analysis

Citrus leaf metabolites were extracted using acidic 80% methanol, as described in our previous studies [17,19,20,115], then derivatized using methyl chloroformate (MCF), as described in our previous studies [17,19,20,115]. After derivatization, 1 μL of the derivatized samples/standards was injected into a GC-MS system model Clarus 680 (Perkin Elmer, Waltham, MA, USA) fitted with ZB-5MS GC column (5% phenyl-arylene, 95% dimethylpolysiloxane; 30 m × 0.25 mm × 0.25 μm film thickness; Phenomenex, Torrance, CA, USA) running in the full scan mode. Helium was used as the carrier gas at a flow rate of 1 mL per minute. The GC thermo program, MS ion identification, data acquisition, and chromatogram analysis were carried out as described in our previous studies [17,20]. Briefly, all detected metabolites were first identified by comparing their mass spectra with spectra from the published literature [17,20]. Subsequently, the identification was verified using library entries of the Wiley Registry of Mass Spectral Data, 9th Edition (John Wiley and Sons, Inc., Hoboken, NJ, USA), NIST 2011 Mass Spectral Library (National Institute of Standards and Technology, Gaithersburg, MA, USA), and the GMD (Golm Metabolome Database; http://gmd.mpimp-golm.mpg.de/, accessed on 20 October 2021). Finally, the identification of all detected metabolites was further confirmed by comparing their retention times (RT), linear retention indices (LRIs), and mass spectra with those of authentic reference standards (Sigma-Aldrich, St. Louis, MO, USA) treated identically to samples. Calibration curves, using a series of reference standards (0, 5, 10, 25, and 50 ppm) derivatized and treated identically to samples, were used for metabolite quantification.

### 4.4. In Situ Histochemical Localization of Hydrogen Peroxide (H_2_O_2_) and Superoxide Anion (O_2_^•−^)

#### 4.4.1. In Situ Histochemical Visualization of O_2_^•−^ Using Nitro Blue Tetrazolium (NBT)

In situ histochemical localization of superoxide anion (O_2_^•−^) was carried out using the nitro blue tetrazolium (NBT; Sigma-Aldrich, St. Louis, MO, USA), as described previously by Romero-Puertas et al. and Shi et al. [116,117], with slight modifications as described in our previous study [118]. Briefly, 20 leaf discs (10 from the lamina and 10 from the midrib, approximately 1 cm^2^ each) were excised, immersed, and vacuum infiltrated in 1 mg mL^−1^ of NBT in 50 mM potassium phosphate buffer (pH 6.4) as described by Ádám et al. [119]. NBT-stained leaves were incubated under light for 20 min at room temperature until the development of a dark blue/purple color characteristic of the reaction of NBT with O_2_^•−^. For better visualization, leaf discs were bleached in 0.15% (*w*/*v*) trichloroacetic acid (TCA) in ethanol:chloroform 4:1 (*v*/*v*). The bleaching solution was exchanged once during the next 24 h of incubation. Then, leaves were stored in 50% glycerol for evaluation. After bleaching, discs were photographed and the intensity of the blue color was quantified using the “Fiji” version of ImageJ (http://fiji.sc, accessed on 14 August 2021) [120].

#### 4.4.2. In Situ Histochemical Localization of H_2_O_2_ Using 3,3′-diamino-benzidine (DAB)

Localization and histochemical staining of H_2_O_2_ was performed using 3,3′-diamino-benzidine (DAB; Sigma-Aldrich, St. Louis, MO, USA) [116,117] with slight modifications, as described in our previous study [118]. Briefly, 20 leaf discs (10 from the lamina and 10 from the midrib, approximately 1 cm^2^ each) were immersed and vacuum infiltrated with 1 mg mL^−1^ of DAB in 10 mM 2-(*N*-morpholino)ethanesulfonic acid (MES) buffer (pH 6.5) and incubated at room temperature under light for 8 h until the development of brown color characteristics of the reaction of DAB with H_2_O_2_. After the illumination, leaf discs were bleached and photographed, and the intensity of the brown color was quantified as described above.

#### 4.4.3. In Situ Fluorescence Localization of Reactive Oxygen Species (ROS) Using H2DCFDA

The generation of ROS was determined using a 2’,7’-dichlorodihydrofluorescein diacetate (H_2_DCFDA) fluorescent probe (emission: 500–560 nm), as described previously [121,122]. Briefly, leaves were excised and vacuum infiltrated for 15 min in a loading buffer (10 mM MES, 50 mM KCl, pH 7.2) with 50 μM H_2_DCFDA, then incubated for 1 h in the dark. After incubation, leaves were washed for 5 min in the loading buffer to remove the excess dye. After washing, leaves were captured using a Canon PowerShotS3 IS (Martin Microscope Co., Easley, SC, USA) coupled with a Wild Heerbrugg stereoscope (Wild Heerbrugg Instruments, Ltd., Heerbrugg, Switzerland) with Stereo Microscope Fluorescence Adapter (SFA-LFS-GO; Nightsea, Lexington, MA, USA) and filter set (royal blue with a green-only bandpass filter; excitation: 440–460 nm, emission: 500–560 nm; model SFA-DL-RB-GO, SunriseDino, Torrance, CA, USA). For high magnification examinations, images were captured with a Zeiss AxioCam ICc 1 Color Microscope Camera (Carl Zeiss Microscopy GmbH, Göttingen, Germany) fitted with AxioScope A1 fluorescent microscope with filter Set 43 or Rhodamine filter (excitation: BP 545/25, emission: BP 525/50) for red and green fluorescence (Carl Zeiss Microscopy GmbH, Göttingen, Germany). Captured high-magnification images were analyzed and the 2D fluorescence images were generated using ZEISS ZEN 3.1 blue edition image processing software (Carl Zeiss Microscopy GmbH, Göttingen, Germany).

### 4.5. Hydrogen Peroxide (H_2_O_2_) Colorimetric Assay

For colorimetric determination of hydrogen peroxide (H_2_O_2_), approximately 0.1 g of ground leaf tissues were extracted on ice using 1 mL of 0.1% (*w*/*v*) TCA as described by Sergiev et al. [123], with slight modifications. After extraction, the homogenate was centrifuged at 1200*× g* for 15 min. H_2_O_2_ content was colorimetrically determined as described by Velikova et al. [124]. Briefly, 100 μL of the supernatant was mixed with 100 μL of 10 mM potassium phosphate buffer (pH 7.0) and 200 μL potassium iodide (1 M) and vortexed for 30 s absorbance was read at 390 nm using a Synergy^TM^ HTX multi-mode microplate reader (BioTek, Winooski, VT, USA). The content of H_2_O_2_ was determined using a calibration curve using H_2_O_2_ solution (30% [*w*/*w*] in H_2_O, Sigma-Aldrich, St. Louis, MO, USA) and expressed as nmol g^−1^ FW.

### 4.6. Gene Expression Analysis

The transcript levels of 67 genes (Appendix A) involved in GABA shunt (18 genes), AA metabolism (12 genes), and phytohormones biosynthesis (37 genes) were determined in leaves of healthy and ‘*Ca*. L. asiaticus’-infected plants (GABA treated vs. non-treated). Samples were analyzed in triplicate for each biological replicate for each treatment. Briefly, the TriZol^®^ reagent (Ambion^®^, Life Technologies, Grand Island, NY, USA) was used to extract the total RNA as described in our previous studies [17,20]. Then, cDNA was synthesized using a SuperScript first-strand synthesis system (Invitrogen) with random hexamer primers as described by the manufacturer’s instructions. qPCR was performed using SYBR Green PCR master mix (Applied Biosystems, Foster City, CA, USA) using ABI 7500 Fast-time PCR System (Applied Biosystems, Foster City, CA, USA). The relative gene expression was determined using the 2^−ΔΔ^*^C^*_T_ method [125]. Elongation factor-1 alpha (*CsEF-1α*) and F-box/kelch-repeat protein (*CsF-box*) were used a housekeeping/reference genes [126,127].

### 4.7. Statistical Analysis

All experiments were designed using a completely randomized design (CRD). Four treatments included healthy (‘*Ca*. L. asiaticus’-free) versus ‘*Ca*. L. asiaticus’-infected citrus plants (GABA-treated vs. non-treated (control)). In all experiments, unless otherwise stated, six biological (six trees) and two technical replicates per treatment were analyzed (*n = 6*). The technical replicates were used only to test the reproducibility and variability of our extraction and derivatization protocols, as well as the reproducibility of the GC-MS instrument, but were not used for statistical analysis to avoid the possibility of pseudo-replication. The normality and homoscedasticity of the data were tested. Data was normally distributed. A two-tailed *t*-test was used for pairwise statistical comparison between every two treatments including healthy versus ‘*Ca*. L. asiaticus’-infected, non-treated versus GABA treated. However, conducting multiple comparisons at once to compare several groups might produce a higher chance of committing a type I error. Therefore, and to control this, we performed a Bonferroni Correction and adjusted the α level to be equal to α_adjusted_ (α_adjusted_ = α/*n*; where α: the original α level [*p* < 0.05], and *n*: the total number of comparisons). Consequently, statistical significance was established as α_adjusted_ ≤ 0.0083 and we only rejected the null hypothesis when the *p*-value was less than this adjusted α level. Furthermore, principal component analysis (PCA) and its associated scatter and loading plots were performed using the data matrix of all individual metabolites singular value decomposition. Likewise, two-way hierarchical cluster analysis (HCA) associated with heatmaps was carried out using the standardized means of all individual metabolites as well as transcript levels of various genes involved in the biosynthesis of different metabolic pathways, and similarities/variations between treatments were presented as a heat map. Distance and linkage of HCA were conducted using Ward’s minimum variance method [128], with 95% confidence between groups from the discriminant function analysis to construct the similarity dendrograms.

## 5. Conclusions

In consolation, non-targeted metabolomic analysis of ‘Valencia’ sweet orange leaves revealed that GABA accumulation controls the cytosolic GABA transit involved in the central C/N metabolism, regulates the proline/glutamate homeostasis, and/or enables the bypass steps in the TCA cycle to secure more energy that is required for plant adaptation and resilience to biotic and abiotic stress. Briefly, GABA accumulation altered the biosynthesis of several metabolic pathways in both healthy and ‘*Ca*. L. asiaticus’-infected plants compared with non-treated control. These metabolic pathways included the PAA, NPAA, PhA, OA, and TCA cycle, as well as the biosynthesis of multiple phytohormones (SA, *t*JA, ABA, auxins, and ET) (Figure 9). In addition to metabolic reprogramming, GABA accumulation plays a key role in the activation of a complex multilayered antioxidant defense system to maintain the redox status within ‘*Ca*. L. asiaticus’-infected plants. This complex antioxidative system consists of two major components: (i) the enzymatic antioxidant defense machinery (six *POXs*, four *SODs*, and *CAT*) serves as the front line in antioxidant defenses, and (ii) the non-enzymatic antioxidant defense machinery (phenolic acids and phenolic amines) that works as a second defense line against ‘*Ca*. L. asiaticus’-induced ROS in citrus infected leaves.

## Figures and Tables

**Figure 1 plants-12-03753-f001:**
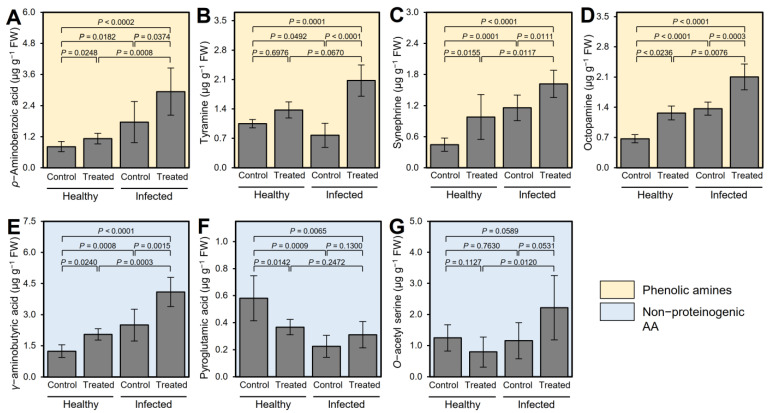
Effect of γ-aminobutyric acid (GABA) supplementation on the endogenous levels of other amines in the leaves of healthy and ‘*Ca*. L. asiaticus’-infected ‘Valencia’ sweet orange (*C. sinensis*) after derivatization with MCF using targeted GC-MS-SIM. (**A**–**D**) Endogenous content (μg g^−1^ FW) of detected phenolic amines (*ρ*-aminobenzoic acid, tyramine, synephrine, and octopamine, respectively) and (**E**–**G**) endogenous content (μg g^−1^ FW) of detected non-proteinogenic amino acids (*γ*-aminobutyric acid (GABA), pyroglutamic acid, and *O*-acetyl serine, respectively) from healthy and ‘*Ca*. L. asiaticus’-infected (infected) leaves without GABA treatment (Control) or after the treatment with 10 mM GABA (treated). Data presented are the means ± standard deviation (mean ± SD) of six biological replicates (*n* = 6). Presented *p*-values are based on a two-tailed *t*-test to statistically compare each pair of treatments. Statistical significance was established as α_adjusted_ ≤ 0.0083 as calculated by Bonferroni Correction.

**Figure 2 plants-12-03753-f002:**
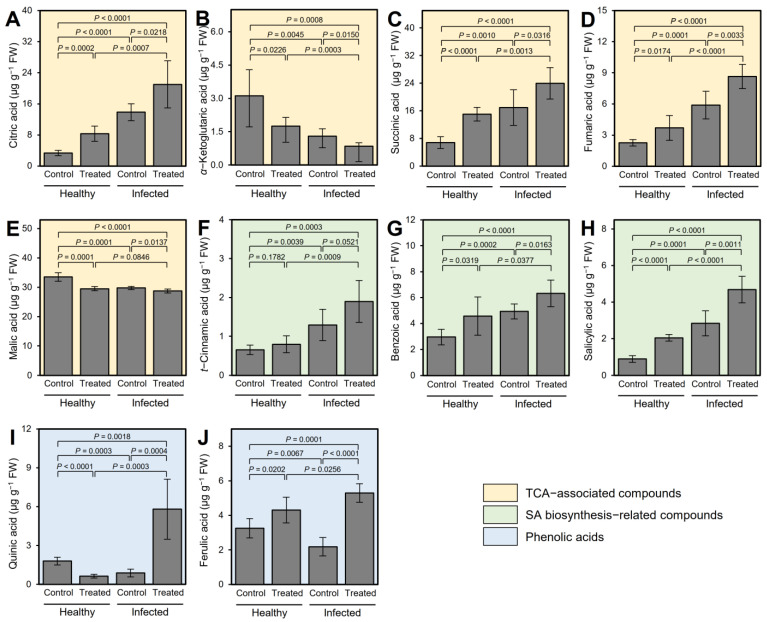
Effect of γ-aminobutyric acid (GABA) supplementation on the endogenous levels of organic acids (OA) in the leaves of healthy and ‘*Ca*. L. asiaticus’-infected ‘Valencia’ sweet orange (*C. sinensis*) after derivatization with MCF using targeted GC-MS-SIM. (**A**–**E**) Endogenous content (μg g^−1^ FW) of detected TCA-associated compounds (citric acid, *α*-ketoglutaric acid, succinic acid, fumaric acid, and malic acid, respectively), (**F**–**H**) SA biosynthesis-related compounds (*t*-cinnamic acid, benzoic acid, and salicylic acid, respectively), and (**I**,**J**) phenolic acids (quinic acid and ferulic acid, respectively) from healthy and ‘*Ca*. L. asiaticus’-infected (infected) leaves without GABA treatment (Control) or after the treatment with 10 mM GABA (treated). Data presented are the means ± standard deviation (mean ± SD) of six biological replicates (*n* = 6). Presented *p*-values are based on a two-tailed *t*-test to statistically compare each pair of treatments. Statistical significance was established as α_adjusted_ ≤ 0.0083 as calculated by Bonferroni Correction.

**Figure 3 plants-12-03753-f003:**
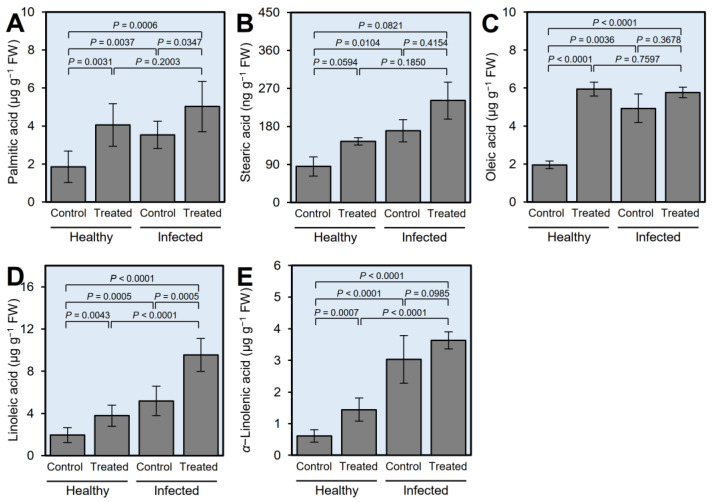
Effect of γ-aminobutyric acid (GABA) supplementation on the endogenous levels of fatty acids in the leaves of healthy and ‘*Ca*. L. asiaticus’-infected ‘Valencia’ sweet orange (*C. sinensis*) after derivatization with MCF using targeted GC-MS-SIM. (**A**–**E**) Endogenous content (μg g^−1^ FW) of detected fatty acids (palmitic acid, stearic acid, oleic acid, linoleic acid, and *α*-linolenic acid, respectively) from healthy and ‘*Ca*. L. asiaticus’-infected (infected) leaves without GABA treatment (Control) or after the treatment with 10 mM GABA (treated). Data presented are the means ± standard deviation (mean ± SD) of six biological replicates (*n* = 6). Presented *p*-values are based on a two-tailed *t*-test to statistically compare each pair of treatments. Statistical significance was established as α_adjusted_ ≤ 0.0083 as calculated by Bonferroni Correction.

**Figure 4 plants-12-03753-f004:**
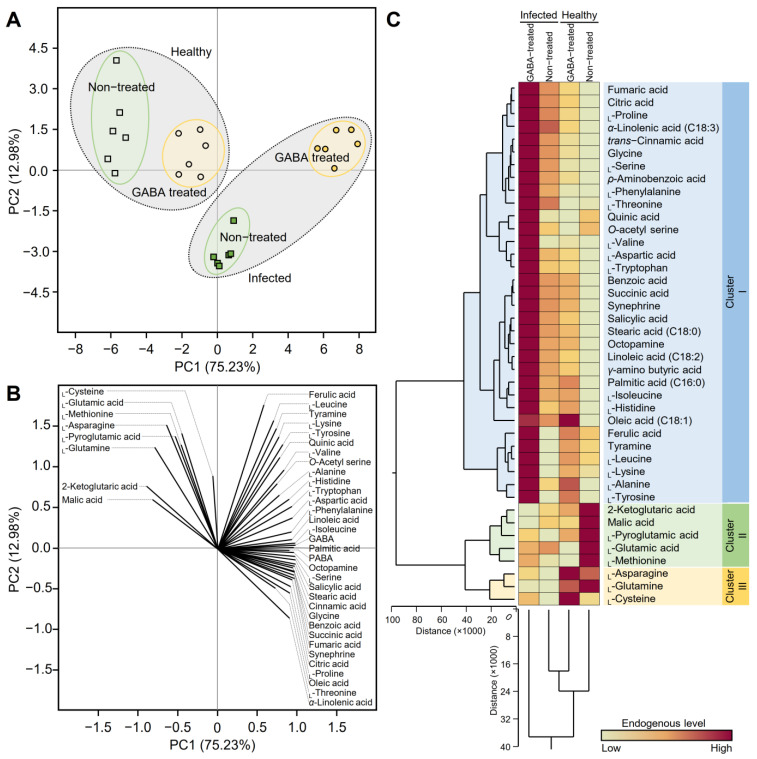
Principal component analysis (PCA) and two-way hierarchical cluster analysis (HCA) of individual metabolites detected in healthy and ‘*Ca*. L. asiaticus’-infected ‘Valencia’ sweet orange (*C. sinensis*) leaves without GABA treatment (Control) or after the treatment with 10 mM GABA (treated). (**A**) PCA-associated scatter plot, (**B**) PCA-associated loading plot, and (**C**) two-way HCA. Variations in metabolite abundances among studied treatments are visualized as a heat map. Rows correspond to individual metabolites, whereas columns correspond to different treatments. Low numerical values are lime green colored, while high numerical values are colored plum red (see the scale at the top right corner of the heat map).

**Figure 5 plants-12-03753-f005:**
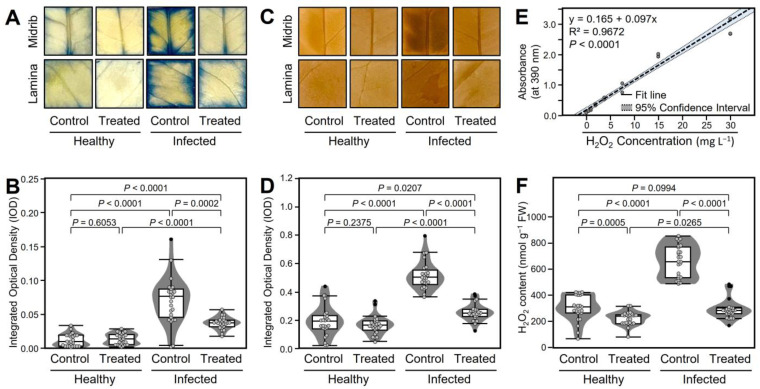
Effect of γ-aminobutyric acid (GABA) supplementation on oxidative stress in the leaves of healthy and ‘*Ca*. L. asiaticus’-infected ‘Valencia’ sweet orange (*C. sinensis*) under greenhouse conditions. (**A**) In situ histochemical visualization of superoxide anion (O_2_*^•^*^−^) using NBT-based staining at 7 dpt with GABA; (**B**) O_2_*^•^*^−^ content based on the NBT staining method at 7 dpt with GABA; (**C**) in situ histochemical visualization of H_2_O_2_ after DAB staining at 7 dpt with GABA; (**D**) H_2_O_2_ content based on the DAB staining method at 7 dpt with GABA; (**E**) calibration curve using different concentrations of H_2_O_2_; and (**F**) H_2_O_2_ content based on the colorimetric method. The minimum and the maximum values are presented by whiskers, while horizontal thick lines specify the median. Boxes show the interquartile ranges (25th to 75th percentile of the data), white dots represent the raw data (*n* = 20), and gray shades represent the corresponding violin plot. Presented *p*-values are based on a two-tailed *t*-test to statistically compare each pair of treatments. Statistical significance was established as α_adjusted_ ≤ 0.0083 as calculated by Bonferroni Correction.

**Figure 6 plants-12-03753-f006:**
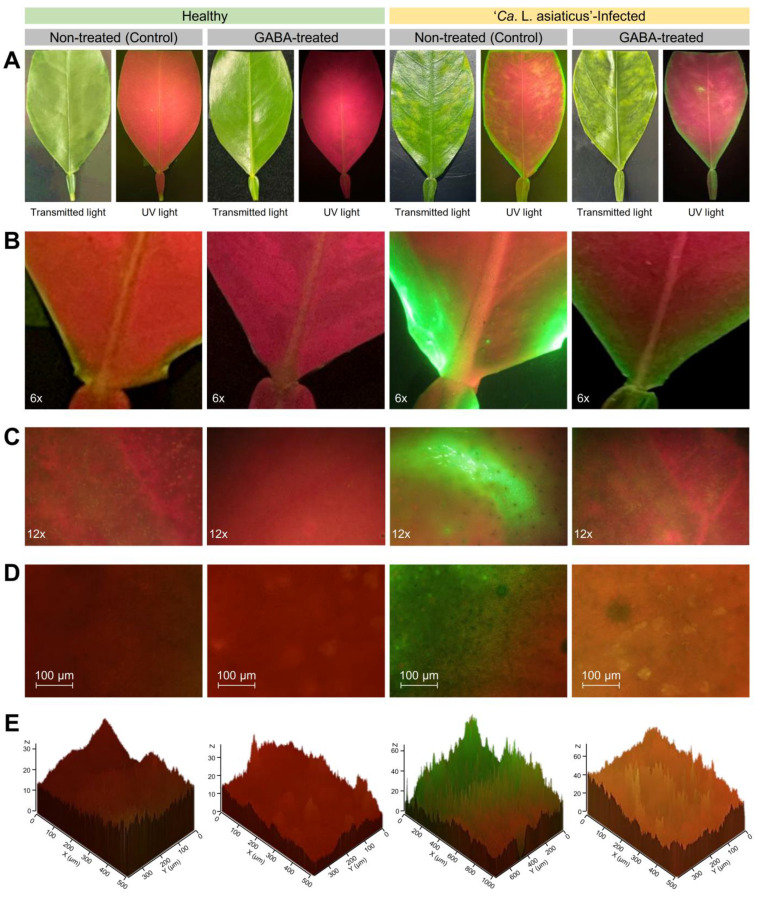
In situ histochemical localization of H_2_O_2_ using H_2_DCFDA-based staining in the leaves of healthy and ‘*Ca*. L. asiaticus’-infected ‘Valencia’ sweet orange (*C. sinensis*) without GABA treatment (Control) or after the treatment with 10 mM GABA (treated) under greenhouse conditions. (**A**) Fluorescence microscopic imaging of H_2_O_2_ localization within the whole leaf under transmitted and UV lights, (**B**) fluorescence microscopic imaging of H_2_O_2_ localization within laminar abscission zone (area between the leaflet blade and petiole wing), (**C**,**D**) fluorescence microscopic imaging of H_2_O_2_ localization within leaf lamina using different magnification, and (**E**) 3D surface plotting of relative fluorescence intensity of non-treated healthy control, GABA-treated healthy control, non-treated ‘*Ca*. L. asiaticus’-infected leaves, and GABA-treated ‘*Ca*. L. asiaticus’-infected leaves, respectively.

**Figure 7 plants-12-03753-f007:**
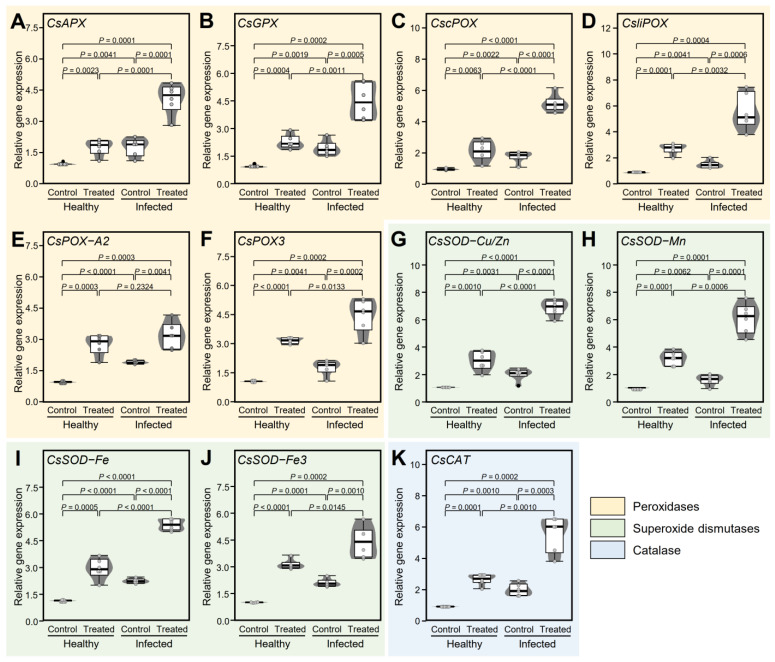
Effect of γ-aminobutyric acid (GABA) supplementation on the relative gene expression of the antioxidant defense-related enzymes in the leaves of healthy and ‘*Ca*. L. asiaticus’-infected ‘Valencia’ sweet orange (*C. sinensis*) without GABA treatment (Control) or after the treatment with 10 mM GABA (treated) under greenhouse conditions. (**A**) Relative gene expression of L-ascorbate peroxidase, cytosolic isoform X1 (*CsAPX*), (**B**) phospholipid hydroperoxide glutathione peroxidase (*CsGPX*), (**C**) cationic peroxidase 1-like (*CscPOX*), (**D**) lignin-forming anionic peroxidase-like (*CsliPOX*), (**E**) peroxidase A2-like (*CsPOX-A2*), (**F**) peroxidase 3 (*CsPOX3*), (**G**) superoxide dismutase (Cu-Zn), chloroplastic (*CsSOD-Cu/Zn*), (**H**) superoxide dismutase (Mn), mitochondrial (*CsSOD-Mn*), (**I**) superoxide dismutase [Fe] 2, chloroplastic-like (*CsSOD-Fe*), (**J**) superoxide dismutase [Fe] 3, chloroplastic isoform X1 (*CsSOD-Fe3*), and (**K**) catalase-like isoform X1 (*CsCAT*). The minimum and the maximum values are presented by whiskers, while horizontal thick lines specify the median. Boxes show the interquartile ranges (25th to 75th percentile of the data), white dots represent the raw data (*n* = 6), and gray shades represent the corresponding violin plot. Presented *p*-values are based on a two-tailed *t*-test to statistically compare each pair of treatments. Statistical significance was established as α_adjusted_ ≤ 0.0083 as calculated by Bonferroni Correction.

**Figure 8 plants-12-03753-f008:**
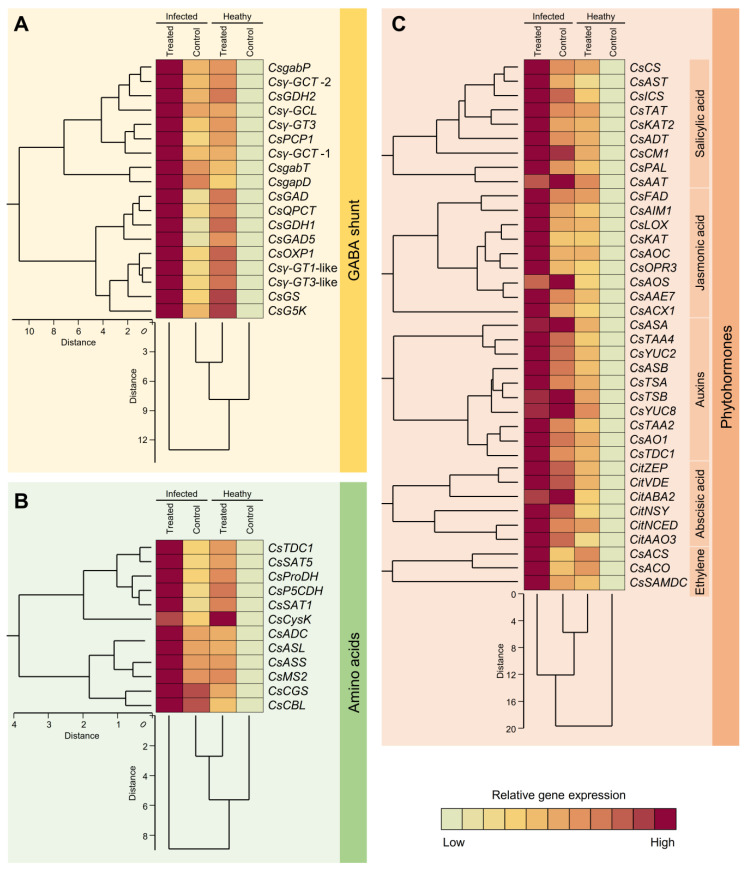
Effect of γ-aminobutyric acid (GABA) supplementation on the relative gene expression of the biosynthetic genes implicated in multiple metabolic pathways in the leaves of healthy and ‘*Ca*. L. asiaticus’-infected ‘Valencia’ sweet orange (*C. sinensis*) without GABA treatment (Control) or after the treatment with 10 mM GABA (treated) under greenhouse conditions. (**A**) Relative gene expression of GABA shunt-related genes, (**B**) amino acids metabolism-related genes, and (**C**) phytohormones biosynthesis-related genes. Variations in relative gene expression among studied treatments are visualized as a heat map. Rows correspond to individual genes, whereas columns correspond to different treatments. Low numerical values are lime green colored, while high numerical values are colored plum red (see the scale at the bottom right corner of the heat map). The full list of expressed genes, names, accession numbers, and primers is available in Appendix A.

**Figure 9 plants-12-03753-f009:**
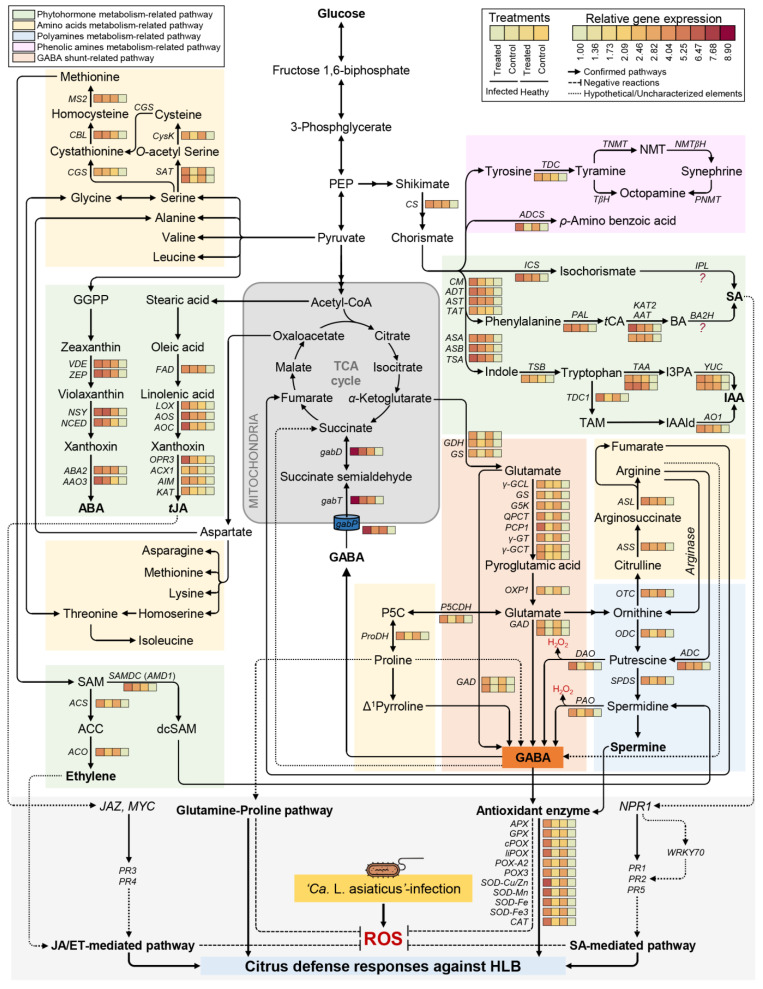
Hypothetical model for the potential effects of γ-aminobutyric acid (GABA) supplementation on different metabolic pathways in the leaves of ‘Valencia’ sweet orange (*C. sinensis*) and their roles in citrus defense responses. In this model, we proposed that exogenous GABA application via root drench might enhance citrus response to the infection with the bacterial pathogen ‘*Ca*. L. asiaticus’ via modulation of multiple metabolic pathways including phytohormone metabolism-related pathways (SA, JA, ABA, ET, and auxins), amino acids metabolism-related pathways, phenolic amines metabolism-related pathway, GABA shunt-related pathway, and TCA-associated compounds. Briefly, (i) exogenous GABA accumulation, due to ‘*Ca*. L. asiaticus’ infection or exogenous GABA application, boosts the accumulation of _L-_tryptophan, _L_-phenylalanine, and α-linolenic acid, the main precursors of auxins, SA, and JA, respectively. Moreover, induced GABA levels upregulated most, if not all, of phytohormones biosynthesis genes in both healthy and ‘*Ca*. L. asiaticus’-infected leaves. (ii) GABA accumulation might boost the plant’s defensive responses against phytopathogenic microbes via augmenting TCA cycle activity. GABA shunt is functionally linked with the TCA cycle via three *gab* genes that enable the non-cyclic flux from *α*-ketoglutarate to succinate in ‘*Ca*. L. asiaticus’-infected citrus plants. (iii) GABA accumulation is also associated with the activation of a complex multilayered antioxidant defense system to maintain the redox status within ‘*Ca*. L. asiaticus’-infected plants. This complex antioxidative system consists of two major components: (i) the enzymatic antioxidant defense machinery (six *POXs*, four *SODs*, and *CAT*) serves as the front line in antioxidant defenses, and (iv) the non-enzymatic antioxidant defense machinery (phenolic acids and phenolic amines) that works as a second defense line against ‘*Ca*. L. asiaticus’-induced ROS in citrus infected leaves. Solid lines with arrows indicate the well-established/confirmed pathways and the dashed lines with whiskers signify negative reactions, whereas round-dotted lines represent hypothetical mechanisms or uncharacterized elements. Relative gene expressions of all studied genes are presented as a heatmap using the non-standardized gene expression patterns. Low numerical values are lime green colored, while high numerical values are colored plum red (see the scale at the top right corner of the heat map). The full list of expressed genes, names, accession numbers, and primers is available in Appendix A.

**Table 1 plants-12-03753-t001:** Concentrations of different amino acids (ng g^−1^ FW) detected in the leaves of healthy and ‘*Ca*. L. asiaticus’-infected ‘Valencia’ sweet orange (*C. sinensis*) after γ-aminobutyric acid (GABA) supplementation.

	Healthy ^y^			‘*Ca*. L. asiaticus’-Infected			*p* Value ^z^		
	Non-Treated (T1)	GABA Treated (T2)	Non-Treated (T3)	GABA Treated (T4)	T1 vs. T2	T1 vs. T3	T1 vs. T4	T2 vs. T4	T3 vs. T4
Glycine	32.8 ± 3.8	37.8 ± 2.2	53.8 ± 8.0	72.6 ± 7.9	0.0196	0.0002	<0.0001	<0.0001	0.0022
_L_-Alanine	770.2 ± 311.2	2023.6 ± 301.4	1177.5 ± 226.3	2271.4 ± 627.8	<0.0001	0.0268	0.0004	0.4039	0.0025
_L_-Valine	185.7 ± 54.5	189.5 ± 25.8	187.1 ± 38.3	265.6 ± 37.1	0.8794	0.9609	0.0141	0.0021	0.0048
_L_-Leucine	70.0 ± 20.0	76.4 ± 14.7	64.8 ± 12.1	87.0 ± 16.2	0.5406	0.5992	0.1374	0.2642	0.0229
_L_-Isoleucine	73.2 ± 14.0	150.0 ± 18.1	142.7 ± 32.1	219.5 ± 83.6	<0.0001	0.0007	0.0018	0.0747	0.062
_L_-Threonine	171.3 ± 23.9	163.5 ± 46.7	318.2 ± 41.4	399.5 ± 67.1	0.721	<0.0001	<0.0001	<0.0001	0.0301
_L_-Asparagine	12,109.1 ± 1636.3	13,754.2 ± 407.6	6075.3 ± 976.2	7920.5 ± 814.5	0.038	<0.0001	0.0002	<0.0001	0.0052
_L_-Proline	16,371.3 ± 1755.6	22,141.5 ± 7199.7	33,562.7 ± 1616.1	43,790.3 ± 9064.0	0.0856	<0.0001	<0.0001	0.001	0.0215
_L_-Aspartic acid	870.8 ± 175.7	927.2 ± 303.1	969.2 ± 279.6	1178.0 ± 393.3	0.7015	0.4824	0.1112	0.2443	0.314
_L_-Serine	3517.1 ± 1058.7	3689.6 ± 621.0	5007.5 ± 2517.4	6734.2 ± 1528.8	0.7377	0.2109	0.0017	0.0011	0.1815
_L_-Glutamine	5548.5 ± 1159.5	4667.8 ± 520.0	1294.6 ± 282.2	1431.8 ± 253.0	0.1204	<0.0001	<0.0001	<0.0001	0.396
_L_-Glutamic acid	3366.5 ± 953.62	2358.81 ± 194.49	2330.5 ± 440.48	2686.36 ± 844.23	0.0967	0.5898	0.4392	0.282	0.8165
_L_-Methionine	129.5 ± 60.5	31.4 ± 12.4	36.6 ± 17.2	64.2 ± 18.0	0.003	0.0047	0.0298	0.0043	0.0214
_L_-Cysteine	26.2 ± 11.5	46.8 ± 16.9	22.8 ± 7.2	29.5 ± 8.2	0.0331	0.5514	0.5802	0.0479	0.1648
_L_-Phenylalanine	326.0 ± 54.9	367.2 ± 67.6	799.9 ± 75.7	1738.7 ± 445.9	0.2733	<0.0001	<0.0001	<0.0001	0.0005
_L_-Lysine	156.2 ± 33.1	211.4 ± 155.9	134.5 ± 28.3	365.1 ± 95.6	0.416	0.2499	0.0005	0.0664	0.0002
_L_-Histidine	2027.6 ± 554.3	2354.7 ± 653.9	2226.9 ± 614.9	2691.2 ± 723.8	0.372	0.5684	0.1049	0.4178	0.2587
_L_-Tyrosine	401.9 ± 152.6	647.5 ± 117.1	396.5 ± 138.8	778.7 ± 251.9	0.0107	0.9502	0.0106	0.274	0.0086
_L_-Tryptophan	62.8 ± 12.5	83.2 ± 14.0	91.1 ± 13.6	180.7 ± 48.5	0.024	0.0038	0.0002	0.0008	0.0014

^y^ Data presented means ± standard deviation (mean ± SD) of six biological replicates. ^z^ Presented *p*-values are based on a two-tailed *t*-test to statistically compare between each pair of treatments. Statistical significance was established as α_adjusted_ ≤ 0.0083 as calculated by Bonferroni Correction.

## Data Availability

All relevant data that support the findings of this study can be found within the article and its Appendix A.

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
