# Peer review of "Gamma-Aminobutyric Acid Accumulation Contributes to Citrus sinensis Response against ‘Candidatus Liberibacter Asiaticus’ via Modulation of Multiple Metabolic Pathways and Redox Status"

_plants, 2023, doi:10.3390/plants12213753_

Round 1
Reviewer 1 Report
Comments and Suggestions for Authors
This paper is well written, and of somewhat interest to readers. The data presented are well analysed via standard methods. With regards to the contributions of Gamma-aminobutyric acid (GABA) to plants etc., there are plenty of papers published, which is a vitamin-like compound and involved in plant growth and development, so it should have some responses towards both biotic and abiotic stress. The authors had published two related papers with magazine MPMI in 2017, 2019, respectively. This manuscript has shown some further progress, which is well presented. I would like to raise a questions towards the title, as you claim in the title that GABA accumulation contributes to citrus defense against Candidatus Liberibacter asiaticum(CLas), we understand that the responses could occurr in Valencia sweet orange caused by GABA, but this cultivar is a susceptible variety to CLas, then more varieties with various responses should be involved for analysis, furthermore, when use the concept of "defense against" , transgenic work also should be addressed. In this case, I would suggest such word should be cautiously used.
Author Response
This paper is well written, and of somewhat interest to readers. The data presented are well analysed via standard methods. With regards to the contributions of Gamma-aminobutyric acid (GABA) to plants etc., there are plenty of papers published, which is a vitamin-like compound and involved in plant growth and development, so it should have some responses towards both biotic and abiotic stress. The authors had published two related papers with magazine MPMI in 2017, 2019, respectively. This manuscript has shown some further progress, which is well presented.
Response: Firstly, thank you very much for your time and efforts in reviewing our manuscript. We appreciate all your comments and suggestions, which enhanced the quality of the manuscript. We have addressed all your comments in the attached file, point-by-point, with no exception. We believe all of those have been addressed in a satisfactory manner.
I would like to raise a questions towards the title, as you claim in the title that GABA accumulation contributes to citrus defense against Candidatus Liberibacter asiaticum(CLas), we understand that the responses could occur in Valencia sweet orange caused by GABA, but this cultivar is a susceptible variety to CLas, then more varieties with various responses should be involved for analysis.
Response: DONE, the title was modified to be more specific and to emphasize the tested cultivar, as follows: “Gamma-aminobutyric acid accumulation contributes to Citrus sinensis response against ‘Candidatus Liberibacter asiaticus’ via modulation of multiple metabolic pathways and redox status”
furthermore, when use the concept of "defense against" , transgenic work also should be addressed. In this case, I would suggest such word should be cautiously used.
Response: DONE, the word “defense” was replaced by the word “response”, as follows: “Gamma-aminobutyric acid accumulation contributes to Citrus sinensis response against ‘Candidatus Liberibacter asiaticus’ via modulation of multiple metabolic pathways and redox status”
Reviewer 2 Report
Comments and Suggestions for Authors
Nahela and Killiny present an elegant manuscript, in which they demonstrate with a robust experimental evidence that GABA application, significantly increased the endogenous levels of several phenolic metabolites in CLas infected leaves. The authors relate this finding with the activation of the non-enzymatic antioxidant defense machinery.
The manuscript is very well written, it is easy to follow and the methodological approaches are adequate enough to support the results and conclusion presented. The tables and figures in manuscript provide information that is easy to understand and are refreshing and visually pleasing.
In my opinion this manuscript presents an eco-friendly alternative that helps citrus trees battle against HLB, and other citrus diseases.
There is jus one concern in discussion section : lines 526-531 In order to clarify the idea about the potential effect of GABA in the posible disruption of the QS system in CLas, despite the fact that the corresponding reference is present in the amnuscript, it would be pertinent to introduce a few lines in which the potential components of the QS system and the routes present in CLas are briefly described
I think that the manuscript contains relevant information about an eco-friendly alternative to the management of the devastating citrus disease HLB, In my opinion it is of major importance for the plant's readers as well as for the several researchers who work in the field of plant-microbe interaction.
Comments on the Quality of English Language
English is accurate and easy to follow
Author Response
Nahela and Killiny present an elegant manuscript, in which they demonstrate with a robust experimental evidence that GABA application, significantly increased the endogenous levels of several phenolic metabolites in CLas infected leaves. The authors relate this finding with the activation of the non-enzymatic antioxidant defense machinery.
The manuscript is very well written, it is easy to follow, and the methodological approaches are adequate enough to support the results and conclusion presented. The tables and figures in manuscript provide information that is easy to understand and are refreshing and visually pleasing.
In my opinion this manuscript presents an eco-friendly alternative that helps citrus trees battle against HLB, and other citrus diseases.
Response: Firstly, thank you very much for your time and efforts in reviewing our manuscript. We appreciate all your comments and suggestions, which enhanced the quality of the manuscript. We have addressed all your comments in the attached file, point-by-point, with no exception. We believe all of those have been addressed in a satisfactory manner.
There is jus one concern in discussion section : lines 526-531 In order to clarify the idea about the potential effect of GABA in the possible disruption of the QS system in CLas, despite the fact that the corresponding reference is present in the manuscript, it would be pertinent to introduce a few lines in which the potential components of the QS system and the routes present in CLas are briefly described
Response: DONE, a new paragraph was written to better discuss the potential role(s) of GABA in the QS system in CLas, as follows:-
“Quorum sensing (QS) is a process of cell-to-cell communication in bacteria that coordinates their behavior and regulates their gene expression based on the population density [77,78]. The LuxR/LuxI-type QS system is one of the most extensively studied QS systems in Gram-negative bacteria [79]. The LuxI protein is an autoinducer synthase that catalyzes the formation of a specific acyl-homoserine lactone (AHL) molecule [79]. The AHL molecule diffuses through the cell membrane at high cell density and binds to the LuxR protein, which is a transcriptional regulator [79]. The LuxR-AHL complex then activates the transcription of its target genes [79]. QS was proposed previously to be crucial for the growth and multiplication of ‘Ca. L. asiaticus’ [80]. It is worth mentioning that the genome of ‘Ca. L. asiaticus’ possesses the LuxR protein, one of a two-component cell-to-cell communication system [81]. For instance, the transcription levels of two putative LuxR family transcriptional regulators were significantly increased in the host plant [80]. However, the genome lacks the second component, LuxI, that produces acyl-homoserine lactones (AHLs), suggesting that ‘Ca. L. asiaticus’ has a solo LuxR system [81]. The absence of LuxI suggests that ‘Ca. L. asiaticus’ may use alternative signaling molecules to regulate gene expression [81]. However, the exact mechanism by which ‘Ca. L. asiaticus’ regulates gene expression in the absence of AHLs is not yet fully understood. It has been suggested that other signaling molecules such as GABA may be involved in regulating virulence genes and biofilm formation in ‘Ca. L. asiaticus’ [78]. We hypothesize that induced GABA levels in infected plants might be for the benefits of the pathogen itself or it is presumably part of citrus's defensive response role against ‘Ca. L. asiaticus’ to interfere with its QS signaling system. Nevertheless, because of ‘Ca. L. asiaticus’ is not cultured yet, neither the exact mechanism by which GABA regulates virulence genes and biofilm formation in ‘Ca. L. asiaticus’ nor the molecular mechanism(s) behind “how does GABA interfere with the QS signaling system?” is yet fully understood and require further research.”
I think that the manuscript contains relevant information about an eco-friendly alternative to the management of the devastating citrus disease HLB, In my opinion it is of major importance for the plant's readers as well as for the several researchers who work in the field of plant-microbe interaction.
Response: Thanks again for your time, and efforts, as well as your nice words.
Reviewer 3 Report
Comments and Suggestions for Authors
The authors report in this manuscript the impact of GABA on the metabolism in citrus. The new knowledge obtained from the study will facilitate to understand the biochemical influences of citrus greening disease, or huanglongbing (HLB), on the citrus plant. The manuscript can be published in the journal. However, due to wrong analyses in some results, the manuscript should be revised, changing the discussion too. One serious problem in this report is the lack of the examination of the validity in parametric analyses. The plots shown in the manuscript suggest that some assumptions for the use of ANOVA seem to be violated. For example, normality may not be supported in Figure 5. Homoscedasticity (homogeneity of variance) may not be assumed in Figure 3. The assumption for the use of parametric analyses should be checked individually and the validity of the data should be described in Materials and Methods. Without the tests, the manuscript cannot be accepted for the publication in the journal. Other points that should be considered are listed below.
L61 Use the common name psyllid, not psylla, just on the one line above.
L110 What are the other hypotheses? Describe the hypotheses tested in the study, one by one.
Table 1 Check the inequality sign. If “>” is used for the probability in Glycine, it means that the significance probability was larger than 0.0001, but not meaning it is still less than 0.05.
L164-169 Phenylalanine was not doubled by GABA treatment: 326.0 to 367.2 in healthy and 799.9 to 1738.7 in infected.
L169-173 Tryptophan was not increased significantly by GABA in healthy: 62.8 to 83.2, not significantly different with “b”.
L184-187 Pyroglutamic acid was significantly affected by GABA, of which influence was interacted with HLB-infection, though.
L186-189 O-acetyl serine was not significantly increased by GABA in infected, with the same sign, “a”.
L265-267 Not all FAs were significantly increased by GABA. Linoleic acid was not significantly different in healthy as indicated with “c” and α-linoleic acid was not in infected with “a”.
L298, 299 Specify the two treatments: healthy and infected. I think these are only groups distinguished in this study, not treated.
Figure 4 Plot also the loadings of the combination of the HLB-infection group and the GABA treatment. This may help understand the results better.
Figure 5 B, D and F These should be drawn as in Figures 1 to 3.
Figure 6E Characters are too small. Change for larger ones.
L395-412 CsliPOX and CxSOD-Mn were not significantly increased by HLB in trees without GABA.
L686 Were these trees produced from seeds, by grafting or by marcotting?
L691 Refer to the date of the inoculation.
L698 How many trees were prepared?
L700 Mention the date of the treatment.
L710 When were the leaves collected from trees tested and how were they selected on the canopy of the trees?
L719 Refer to the date of the extraction.
L767- Specify the wave lengths examined.
L774 Refer to the wave length of the filter.
L797 What does “biological replicate” mean?
L807- I think that at least some data should have been transformed appropriately for parametric analyses. Explain the validity of the data for these analyses.
Refer to the distance and method for the cluster analyses. Different distance measures and/or methods may give different results.
Comments on the Quality of English LanguageThe manuscript is well written in English. However, the use of "believe" may confuse readers. Since this is a scientific report, this word should be avoided.
Author Response
The authors report in this manuscript the impact of GABA on the metabolism in citrus. The new knowledge obtained from the study will facilitate to understand the biochemical influences of citrus greening disease, or huanglongbing (HLB), on the citrus plant. The manuscript can be published in the journal.
Response: Firstly, thank you very much for your time and efforts in reviewing our manuscript. We appreciate all your comments and suggestions, which enhanced the quality of the manuscript. We have addressed all your comments in the attached file, point-by-point, with no exception. We believe all of those have been addressed in a satisfactory manner.
However, due to wrong analyses in some results, the manuscript should be revised, changing the discussion too. One serious problem in this report is the lack of the examination of the validity in parametric analyses. The plots shown in the manuscript suggest that some assumptions for the use of ANOVA seem to be violated. For example, normality may not be supported in Figure 5. Homoscedasticity (homogeneity of variance) may not be assumed in Figure 3. The assumption for the use of parametric analyses should be checked individually and the validity of the data should be described in Materials and Methods. Without the tests, the manuscript cannot be accepted for the publication in the journal.
Response: DONE, Firstly, we would like to thank the reviewer for pointing out this issue. A paired t-test (as a parametric test) was used for pairwise statistical comparison between each two treatments including healthy versus ‘Ca. L. asiaticus’-infected, non-treated versus GABA-treated, and statistical significance was established as p < 0.05. Accordingly, most of the figures were reconstructed and their associated results were rewritten. Moreover, the normality and homoscedasticity of the data were tested. Data was normally distributed.
Other points that should be considered are listed below.
L61 Use the common name psyllid, not psylla, just on the one line above.
Response: DONE, the word “psylla” was replaced with “psyllid”
L110 What are the other hypotheses? Describe the hypotheses tested in the study, one by one.
Response: DONE, the sentence was rewritten to ensure clarity, as follows: ‘Moreover, GABA-mediated defense responses are associated with the reduction of reactive oxygen species (ROS) accumulation via activation of antioxidant defense machinery”
Table 1 Check the inequality sign. If “>” is used for the probability in Glycine, it means that the significance probability was larger than 0.0001, but not meaning it is still less than 0.05.
Response: DONE, we agree with the reviewer. Sorry, it was a typo mistake. All “greater than” signs (>) were replaced with “less than” signs (<) when used for the probability. This issue has been revised throughout the manuscript and corrected wherever it was needed.
L164-169 Phenylalanine was not doubled by GABA treatment: 326.0 to 367.2 in healthy and 799.9 to 1738.7 in infected.
Response: DONE, the word “doubled” was replaced with the word “increased”
L169-173 Tryptophan was not increased significantly by GABA in healthy: 62.8 to 83.2, not significantly different with “b”.
Response: DONE, the sentence was rewritten to ensure clarity. However, based on the new statistical analysis that we did in the revised version of this manuscript, “L-tryptophan, was similarly heightened in ‘Ca. L. asiaticus’-infected leaves (91.1±13.6 ng g-1 FW) compared to non-treated healthy control (62.8±12.5 ng g-1 FW), however, its levels were also increased upon GABA application in both healthy (83.2±14.0 ng g-1 FW) and treated plants (180.7±48.5 ng g-1 FW)”
|
Healthy y |
‘Ca. L. asiaticus’-infected |
P value z |
|||||||
|
|
Nontreated (T1) |
GABA-treated (T2) |
Nontreated (T3) |
GABA-treated (T4) |
T1 vs. T2 |
T1 vs. T3 |
T1 vs. T4 |
T2 vs. T4 |
T3 vs. T4 |
|
L-Tryptophan |
62.8 ± 12.5 |
83.2 ± 14.0 |
91.1 ± 13.6 |
180.7 ± 48.5 |
0.0242 |
0.0039 |
0.0014 |
0.0035 |
0.0052 |
L184-187 Pyroglutamic acid was significantly affected by GABA, of which influence was interacted with HLB-infection, though.
Response: DONE, because all other amines were changed upward, but Pyroglutamic acid changed downward, the section 2.2. title was changed to “2.2. GABA accumulation alters the endogenous levels of other amines in both healthy and ‘Ca. L. asiaticus’-infected plants”.
Moreover, the description of figure 1F was rewritten to ensure clarity, as follows: “On the other hand, ‘Ca. L. asiaticus’ infection significantly reduced the levels of pyroglutamic acid in nontreated leaves compared with non-treated control (p = 0.0020; Figure 1F). likewise, exogenous GABA application significantly decreased pyroglutamic acid levels in GABA-treated healthy plants (p = 0.0245), but not ‘Ca. L. asiaticus’-infected ones (p = 0.1310; Figure 1F)”
L186-189 O-acetyl serine was not significantly increased by GABA in infected, with the same sign, “a”.
Response: DONE, the sentence was rewritten based on the new statistical analysis that we did in the revised version of this manuscript as follows: “Nevertheless, neither GABA application nor ‘Ca. L. asiaticus’ infection affected the O-acetyl serine content in both healthy and infected citrus plants (Figure 1G)”
L265-267 Not all FAs were significantly increased by GABA. Linoleic acid was not significantly different in healthy as indicated with “c” and α-linoleic acid was not in infected with “a”.
Response: DONE, the paragraph was rewritten according to the new statistical analysis that we did in the revised version of this manuscript, as follows: “The levels of all detected FA were significantly increased in ‘Ca. L. asiaticus’-infected leaves compared to non-treated healthy control. However, root drench application of 10 mM GABA noticeably enhanced the FA profile of all detected compounds in both healthy and infected citrus trees, except for oleic acid and α-linolenic acid in infected leaves. It is worth mentioning that α-linolenic acid (C18:3; the precursor of jasmonic acid [JA]) was significantly heightened in ‘Ca. L. asiaticus’-infected plants (3.04±0.75 µg g-1 FW) compared to non-treated control (0.61±0.20 µg g-1 FW). Nevertheless, GABA supplementation stimulated the accumulation of α-linolenic acid in treated healthy plants (1.45±0.37 ng g-1 FW) compared with non-treated healthy control (p = 0.0014), but not ‘Ca. L. asiaticus’-infected ones (3.63±0.28 ng g-1 FW) when compared with non-treated infected ones (p = 0.1159; Figure 3E)”
L298, 299 Specify the two treatments: healthy and infected. I think these are only groups distinguished in this study, not treated.
Response: DONE, the statement was rewritten to ensure clarity, as follows: “Additionally, the PCA-associated loading plot showed that while eight metabolites were positively correlated with healthy plants (either treated or not), 17 metabolites were associated with GABA-treated, ‘Ca. L. asiaticus’-infected leaves, while the rest (16 compounds) were associated with nontreated ‘Ca. L. asiaticus’-infected plants”
Figure 4 Plot also the loadings of the combination of the HLB-infection group and the GABA treatment. This may help understand the results better.
Response: We agree with the reviewer that plotting the loadings of the combination of the HLB-infection group and the GABA treatment might add more information to this section. However, to keep the simplicity of the figures, we believe that the presented loading plot delivers our idea.
Figure 5 B, D and F These should be drawn as in Figures 1 to 3.
Response: In figures 5B, 5D, and 5F, we prefer to use the boxplot rather than the bar graph to visualize our data, because it is more suited for showing the distribution of continuous, univariate data, by visualizing the following six characteristics of a dataset: minimum, first quartile, median, third quartile, maximum, and outliers, particularly with high numbers of replicates (n = 20).
Figure 6E Characters are too small. Change for larger ones.
Response: DONE, the whole figure was reconstructed to ensure clarity and readability.
L395-412 CsliPOX and CxSOD-Mn were not significantly increased by HLB in trees without GABA.
Response: DONE, according to the new statistical analysis that we did in the revised version of this manuscript, both CsliPOX and CsSOD-Mn were significantly increased in citrus leaves due to the infection with ‘Ca. L. asiaticus’ (p=0.0041 and 0.0062, respectively). The whole paragraph was rewritten to ensure clarity.
L686 Were these trees produced from seeds, by grafting or by marcotting?
Response: DONE, All plant materials were produced by grafting. This issue has been clarified within section “4.1. Plant Materials and Growth Conditions”
L691 Refer to the date of the inoculation.
Response: DONE, citrus plants were inoculated via bud-grafting using ‘Ca. L. asiaticus’-positive materials in March 2020. This issue has been clarified within section “4.1. Plant Materials and Growth Conditions”
L698 How many trees were prepared?
Response: DONE, actually, we prepare a huge number ‘Ca. L. asiaticus’-infected plants regularly in our greenhouse to be ready for further experiments. In our experiment, unless otherwise stated, six biological and two technical replicates per treatment were analyzed (n=6). The technical replicates were used only to test the reproducibility and variability of our extraction and derivatization protocols, as well as the reproducibility of the GC-MS instrument, but were not used for statistical analysis to avoid the possibility of pseudo-replication. This issue has been clarified in section “4.7. Statistical analysis”
L700 Mention the date of the treatment.
Response: DONE, All plant materials (healthy vs. infected and GABA-treated vs. non-treated) were treated in October 2020. This issue has been clarified in section “4.2. Treatment of Citrus Plants with Exogenous GABA and leaf sampling”
L710 When were the leaves collected from trees tested and how were they selected on the canopy of the trees?
Response: DONE, Our previous studies showed that the positive effects of exogenous GABA application reached their highest peak at 7 days post-treatment (dpt). Therefore, all plant materials for all below analyses were collected at 7-dpt. Kindly see :-
- Hijaz, F.; Nehela, Y.; Killiny, N. Application of Gamma-Aminobutyric Acid Increased the Level of Phytohormones in Citrus Sinensis. Planta 2018, 248, 909–918, doi:10.1007/s00425-018-2947-1.
- Hijaz, F.; Killiny, N. The Use of Deuterium-Labeled Gamma-Aminobutyric (D6-GABA) to Study Uptake, Translocation, and Metabolism of Exogenous GABA in Plants. Plant Methods 2020, 16, doi:10.1186/s13007-020-00574-9.
For sampling, three leaves were collected from each biological sample, from different positions and different ages; juvenile leaves from the top, intermediate-aged leaves (fully expanded, but not hardened) from the middle, and mature leaves (deep green and hard-ened) from the lower part of the plant.
This issue has been clarified in section “4.2. Treatment of Citrus Plants with Exogenous GABA and leaf sampling”
L719 Refer to the date of the extraction.
Response: DONE, Citrus leaf metabolites were extracted using acidic 80% methanol during November 2020, however, we believe that this information is not necessary to be mentioned, particularly since it is a lab work. In our lab, usually, the collected samples are homogenized and immediately kept at -80 °C until further analysis.
L767- Specify the wave lengths examined.
Response: DONE, 500-560nm. This issue has been clarified in section “4.4.3. In situ fluorescence localization of reactive oxygen species (ROS) using H2DCFDA”
L774 Refer to the wave length of the filter.
Response: DONE, the used filter set (Royal blue with a green-only bandpass filter; Excitation: 440-460nm, Emission: 500-560nm; model SFA-DL-RB-GO, SunriseDino, Torrance, CA, USA). This issue has been clarified in section “4.4.3. In situ fluorescence localization of reactive oxygen species (ROS) using H2DCFDA”
L797 What does “biological replicate” mean?
Response: DONE, In our experiment, six biological and two technical replicates per treatment were analyzed (n=6). The technical replicates were used only to test the reproducibility and variability of our extraction and derivatization protocols, as well as the reproducibility of the GC-MS instrument, but were not used for statistical analysis to avoid the possibility of pseudo-replication. This issue has been clarified in section “4.7. Statistical analysis”.
In other words, for each treatment we used six replicates (aka biological replicate) with at least five trees per replicates. However, when we run our sample, each sample ran twice (aka technical replicates)
L807- I think that at least some data should have been transformed appropriately for parametric analyses. Explain the validity of the data for these analyses.
Response: DONE, we would like to thank the reviewer for pointing out this issue. A paired t-test (as a parametric test) was used for pairwise statistical comparison between each two treatments including healthy versus ‘Ca. L. asiaticus’-infected, non-treated versus GABA-treated, and statistical significance was established as p < 0.05. Accordingly, most of the figures were reconstructed and their associated results were rewritten. Moreover, the normality and homoscedasticity of the data were tested. Data was normally distributed. This issue has been clarified in section “4.7. Statistical analysis”.
Refer to the distance and method for the cluster analyses. Different distance measures and/or methods may give different results.
Response: DONE, Distance, and linkage of HCA were conducted using Ward’s minimum variance method with 95% confidence between groups from the discriminant function analysis, to construct the similarity dendrograms. This issue has been clarified in section “4.7. Statistical analysis”.
- Ward, J.H. Hierarchical Grouping to Optimize an Objective Function. J. Am. Stat. Assoc. 1963, 58, 236–244.
Comments on the Quality of English Language
The manuscript is well-written in English. However, the use of "believe" may confuse readers. Since this is a scientific report, this word should be avoided.
Response: DONE, the word “believe” was replaced with another appropriate verb based on the context. This issue has been revised and corrected throughout the manuscript.
Round 2
Reviewer 3 Report
Comments and Suggestions for Authors
Thank you for revising the manuscript. Most replies are understood. However, unfortunately, there are fatal mis-usings of statistics in analysing the results. As the authors reply, they used paired t-test to compare the effects on chemical properties among treatments with a significance level at P < 0.05. This must be avoided. If each pair for two treatments in a multi-treatment experiment is compared by a paired t-test, Type I errors are likely led. There are some measures to avoid this problem. Here, two major ones are provided here. The probability may be corrected by some measures. Bonferroni's correction may be used. The other is to apply multi-comparison tests such as Tukey's test or any others. The authors should check the data distribution again and analyse their data by using appropriate measures for the comparisons of the treatment effects. I think the results that will be obtained by any measures may be changed as the statistical analyses and, accordingly, the authors will be required to consider the conclusion from the analyses again.
Author Response
Thank you for revising the manuscript. Most replies are understood.
Response: Firstly, thank you very much for your time and efforts in reviewing our manuscript. We appreciate all your comments and suggestions, which enhanced the quality of the manuscript. We have addressed all your comments in the attached file, point-by-point, with no exception. We believe all of those have been addressed in a satisfactory manner.
However, unfortunately, there are fatal mis-usings of statistics in analyzing the results. As the authors reply, they used paired t-test to compare the effects on chemical properties among treatments with a significance level at P < 0.05. This must be avoided. If each pair for two treatments in a multi-treatment experiment is compared by a paired t-test, Type I errors are likely led. There are some measures to avoid this problem. Here, two major ones are provided here. The probability may be corrected by some measures. Bonferroni's correction may be used. The other is to apply multi-comparison tests such as Tukey's test or any others. The authors should check the data distribution again and analyse their data by using appropriate measures for the comparisons of the treatment effects. I think the results that will be obtained by any measures may be changed as the statistical analyses and, accordingly, the authors will be required to consider the conclusion from the analyses again.
Response: DONE, we agree with the reviewer that conducting multiple comparisons at once to compare several groups might produce a higher chance of committing a type I error. Therefore, and to control this, we performed a Bonferroni Correction and adjusted the α level to be equal to αadjusted (αadjusted= α/n; where: α: the original α level [p < 0.05], and n: the total number of comparisons). Consequently, statistical significance was established as αadjusted < 0.0083 and we only rejected the null hypothesis when the p-value was less than this adjusted α level. Accordingly, several paragraphs within the “Results” section, were rewritten. However, results that changed as the statistical analyses using Bonferroni Correction used, did not affect our “conclusion” significantly.
Fortunately, our discussion and conclusion were built on multiple phytohormone biosynthesis pathways including such as auxins, SA, and JA, as well as both enzymatic and non-enzymatic antioxidant machinery. Briefly, our findings showed that ‘Ca. L. asiaticus’ infection and/or exogenous GABA application resulted in the induction of L-tryptophan, L-phenylalanine, and some fatty acids the main precursors of auxins, SA, and JA, respectively. These main findings did not change after using αadjusted based on Bonferroni Correction.
Thanks for your suggestion, it was very helpful.
Round 3
Reviewer 3 Report
Comments and Suggestions for Authors
Now the manuscript can be accepted for the publication in the journal.
Author Response
Thank you so much.